# Gastrointestinal microbiota composition predicts peripheral inflammatory state during treatment of human tuberculosis

Matthew F. Wipperman [1,2,11], Shakti K. Bhattarai[3,11], Charles Kyriakos Vorkas [1,4], Venkata Suhas Maringati[3], Ying Taur [5], Laurent Mathurin[6], Katherine McAulay[7], Stalz Charles Vilbrun[6], Daphie Francois[6], James Bean[1], Kathleen F. Walsh [7], Carl Nathan[8], Daniel W. Fitzgerald[7], Michael S. Glickman [1,4,8 ✉] & Vanni Bucci [3,9,10 ✉]

The composition of the gastrointestinal microbiota influences systemic immune responses, but how this affects infectious disease pathogenesis and antibiotic therapy outcome is poorly understood. This question is rarely examined in humans due to the difficulty in dissociating the immunologic effects of antibiotic-induced pathogen clearance and microbiome alteration. Here, we analyze data from two longitudinal studies of tuberculosis (TB) therapy (35 and 20 individuals) and a cross sectional study from 55 healthy controls, in which we collected fecal samples (for microbiome analysis), sputum (for determination of *Mycobacterium tuberculosis* (*Mtb*) bacterial load), and peripheral blood (for transcriptomic analysis). We decouple microbiome effects from pathogen sterilization by comparing standard TB therapy with an experimental TB treatment that did not reduce *Mtb* bacterial load. Random forest regression to the microbiome-transcriptome-sputum data from the two longitudinal datasets reveals that renormalization of the TB inflammatory state is associated with *Mtb* pathogen clearance, increased abundance of Clusters IV and XIVa Clostridia, and decreased abundance of Bacilli and Proteobacteria. We find similar associations when applying machine learning to peripheral gene expression and microbiota profiling in the independent cohort of healthy individuals. Our findings indicate that antibiotic-induced reduction in pathogen burden and changes in the microbiome are independently associated with treatment-induced changes of the inflammatory response of active TB, and the response to antibiotic therapy may be a combined effect of pathogen killing and microbiome driven immunomodulation.

[1] Immunology Program, Memorial Sloan Kettering Cancer Center, New York, NY, USA. [2] Clinical and Translational Science Center, Weill Cornell Medicine, New York, NY, USA. [3] Department of Microbiology and Physiological Systems, University of Massachusetts Medical School, Worcester, MA, USA. [4] Division of Infectious Diseases, Weill Cornell Medicine, New York, NY, USA. [5] Division of Infectious Diseases, Memorial Sloan Kettering Cancer Center, New York, NY, USA. [6] Haitian Study Group for Kaposi's Sarcoma and Opportunistic Infections (GHESKIO), Port-au-Prince, Haiti. [7] Center for Global Health, Weill Cornell Medicine, New York, NY, USA. [8] Immunology and Microbial Pathogenesis Graduate Program, Weill Cornell Graduate School, New York, NY, USA. [9] Center for Microbiome Research, University of Massachusetts Medical School, Worcester, MA, USA. [10] Program in Systems Biology, University of Massachusetts Medical School, Worcester, MA, USA. [11] These authors contributed equally: Matthew F. Wipperman, Shakti K. Bhattarai. ✉email: glickmam@mskcc.org; vanni.bucci2@umassmed.edu

There is mounting evidence that the gut microbiome has an important role in the modulation of host physiology, with a wealth of studies having associated microbiome composition and functions with differential inflammatory, neurological, and even behavioral activity[1]. Gastrointestinal colonization by specific taxa with particular metabolic capacities has been shown to differentially modulate host biology[2]. For example, colonization by a subset of Clostridia enhanced anti-inflammatory phenotypes in mice[3], and enrichment in specific members of the *Bacteroides* and *Parabacteroides* genera induced CD8+ T cell responses and anticancer activity in mice and marmosets[4], as well as correlating with the abundance of these immune effectors in humans[5]. A multitude of experiments in mice have allowed for the determination of mechanisms by which gastrointestinal mucosal-associated bacteria affect host physiology at the epithelial interface and systemically throughout their host[6,7].

Despite these observations, it is unknown whether, and to what degree, microbiome changes are associated with systemic changes in inflammatory responses in humans. This knowledge gap is due in part to the difficulty of isolating the microbiome dependent effects from other aspects of human physiology and in discerning the direction of causality in human studies. As microbial communities in the gut promote the development and maintenance of innate and adaptive immune responses[8], including microbiota-educated immune cells and many small molecules that circulate throughout the periphery[9], we would expect to observe both localized and systemic host effects upon major microbiome alterations such as treatment with antibiotics, which we can measure using technologies such as shotgun RNA sequencing (RNAseq)[10] of peripheral blood.

Individuals infected by *Mycobacterium tuberculosis* (*Mtb*) and having active TB disease (the 9th leading cause of death on Earth[11]) has been shown to have a markedly different systemic gene expression profiles compared to people with latent disease, other respiratory diseases, or no known infection[11-13]. Specifically, infection with *Mtb* leads to heightened expression of inflammatory pathways, most notably the Type I and Type II interferon pathways[14-17], with this pattern resolving with antibiotic therapy[14,17,18]. A recent meta-analysis combining microarray and RNAseq data from studies aimed at identifying active TB transcriptional signatures, confirmed the findings about a specific set of peripheral blood transcripts that are biomarkers of active TB disease, relative to healthy individuals or those with latent TB infection (LTBI)[19]. Antibiotic treatment for active TB involves combination therapy with narrow (HZE), and semi-broad (R) spectrum, and prodrug (HZ) agents with mostly *Mycobacterial*-specific targets. HRZE is given for two months and is then followed by an HR-only administration for an additional four months, in order to achieve over 95% likelihood of cure[20]. The disruptive effect of HRZ(E) therapy on the intestinal microbiome was demonstrated in a longitudinal study in mice[21] and cross-sectional study in humans[22], which indicated that the major phyla perturbed are from the class Clostridia, a group of obligate anaerobes in the gut with well described immunomodulatory effects on the host[2,3,23,24].

Given that HRZE treatment causes GI microbiota shifts that include the depletion of many Clostridia species, and given the role that these species play in modulation of host biology in mice and humans, we reasoned that there could be a connection between the microbiome alterations observed during HRZE therapy and the resolution of systemic inflammatory responses to TB. However, because HRZE therapy rapidly reduces the burden of *Mtb* in the early phase of treatment, it is difficult to uncouple the immunologic effects of pathogen killing from microbiota perturbation without a control group that has either pathogen killing or microbiome perturbation, but not both.

To address these questions, we combined three independent clinical datasets for which we had gathered microbiome profiling via 16S ribosomal DNA (rDNA) sequencing, peripheral blood transcriptomics, and *Mtb* abundance in the sputum (in 2 of 3 datasets). The first dataset (Fig. 1A, B) consists of the secondary endpoint data from a longitudinal and interventional clinical trial (NCT02684240) that compared the early bactericidal effect (EBA) of standard tuberculosis (TB) therapy isoniazid (H), rifampin (R), pyrazinamide (Z), and ethambutol (E) (HRZE, arm 1) to the antiparasitic drug nitazoxanide (NTZ, arm 2), shown to possess antimycobacterial activity in vitro[25,26]. The analyzed data included the 29 subjects in the trial, whose primary endpoint results along with randomization procedures, statistical plan, and other details, were recently reported in Walsh et al.[26], and six additional individuals that gave informed consent for the pilot pre-randomization phase of the study, which was run to verify assays and collection protocols (leading to a total $N = 35$ for Cohort 1). We note that compared to Walsh et al.[26] in where 20 people received NTZ we were able to collect stool and blood specimen for only 19 of them to be used in this study. The second dataset (Cohort 2, $N = 20$ at baseline) consisted of an independent 6-month longitudinal and observational HRZE treatment cohort (Fig. 1A, B). We used these two Cohorts to first characterize short (for both HRZE and NTZ treatments) and long-term (for HRZE only) effects on the gastrointestinal microbiota and peripheral gene expression. More importantly we used these two cohorts' data to answer our underlying questions by training Random Forest Regression models to assess the changes in expression of peripheral inflammatory pathways as a function of changes in microbiota species abundances and simultaneous changes in *Mtb* in the sputum (Fig. 1C). Finally, we validated the determined microbiota-peripheral-gene expression relationships using a cross-sectional and observational cohort of healthy Haitian community controls (CC) and healthy household contacts (or Family Contacts, FC) of TB patients (for a total $N = 55$), some previously described[5] (Fig. 1A, B).

## Results

### Gut microbiome diversity is depleted after two weeks of HRZE or NTZ treatment

As detailed elsewhere, the GHESKIO centers in Port au Prince, Haiti conducted a prospective, randomized, early bactericidal activity (EBA) study in treatment-naive, drug-susceptible adult patients with uncomplicated pulmonary tuberculosis (TB) (ClinicalTrials.gov Identifier: NCT02684240)[26]. 30 participants were randomized to receive either NTZ, 1000 mg po (oral) twice daily, or standard oral therapy with isoniazid 300 mg daily, rifampin 600 mg daily, pyrazinamide 25 mg/kg daily, and ethambutol 15 mg/kg daily (referred to as HRZE) for 14 days (Figs. 1A, 2A), and 5 participants were prescreened to receive HRZE. These five additional individuals had given informed consent for the pilot pre-randomization phase of the study (see above) The primary endpoint of the trial was sputum bacterial load (measured by time to culture positivity, TTP) in a BACTEC liquid culture system, as a quantitative microbiologic measure of disease resolution. Sputum was collected from 6 p.m. to 9 a.m. every other day to quantify mycobactericidal activity of each treatment regimen.

HRZE resulted in a predictable increase in the TTP (corresponding to reduced bacterial load) over the first two weeks of therapy compared to baseline TTP ($p < 0.001$) for the linear mixed-effect model $TTP \sim Sex + Age + Time + Treatment + Time : Treatment + 1|ID$ where $Time: Treatment$ is the interaction term and $1|ID$ is the subject-level random effect. NTZ was used as the reference level for treatment (see Methods and Supplementary Data 2 for exact $p$-values) (Fig. 2B). NTZ, despite its potent in vitro activity[27], did not have any significant effect on TTP after 14 days ($p > 0.05$) (Fig. 2B, Supplementary Data 2)[26]. This lack of NTZ antimicrobial effect was traced to a failure of the

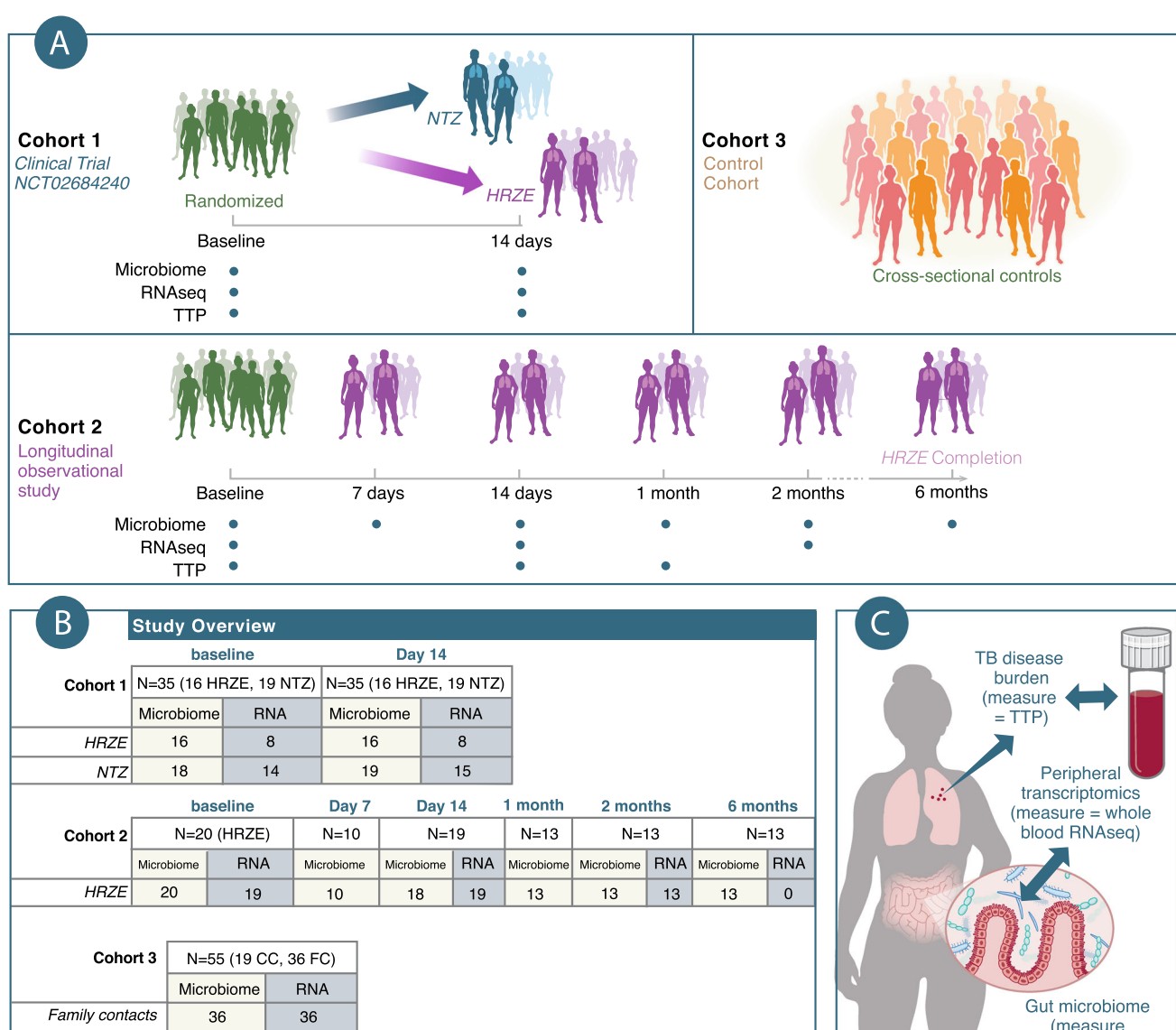

**Fig. 1 Overview of cohorts, subjects, timepoints, samples, and hypotheses in this study. A** This study investigates microbiome-transcriptome relationships in three separate cohorts of individuals in Haiti. Cohort 1 (2-week longitudinal and interventional clinical trial) consists of secondary analysis of a randomized clinical trial of study volunteers, where we collected disease severity measurements (*Mtb* bacterial load, TTP), microbiome profiling, and peripheral transcriptomics in active TB patients at baseline, before randomization to either HRZE (arm 1) (standard of care TB treatment), or Nitazoxanide (NTZ) (arm 2). Cohort 2 (6 month longitudinal and observational study) consists of study volunteers who were followed throughout the course of 6 months of TB treatment, where we collected TTP, microbiome, and transcriptomics data. Finally, Cohort 3 (cross sectional and observational) consists of healthy volunteers. These healthy volunteers were enrolled separately. Around half are healthy and TB-negative household or family contacts (FC) of active TB-patients, and the other half are community controls (CC), with no know TB exposure. We performed microbiome profiling and peripheral transcriptomics on these individuals as well. **B** Numbers of individuals in this study. **C** Diagram showing the major questions investigated in this study. Supplementary Data 1 provides a table with dates of first and last enrollment for every Cohort.

drug to penetrate the sputum[26]. All patients were subsequently switched to HRZE standard of care treatment.

We have previously reported[22] that HRZE therapy depletes members of the order Clostridiales, but the cross-sectional design of that study did not allow for conclusions about the rapidity of this effect, and most importantly, did not include pretreatment samples to allow for the assessment of baseline microbiome composition. To investigate microbiome changes induced by HRZE or NTZ, we extracted and amplified bacterial and archaeal DNA using V4–V5 16S rDNA sequencing (see "Methods" section). Stool samples were collected at baseline before the start of treatment and on day 14 of therapy (Fig. 1A). Using Principal Coordinate Analysis (PCoA)

with Bray–Curtis distances, we found that that the component accounting for the greatest variation in the microbiome data qualitatively represented changes in microbiome community structure that occur after two weeks of NTZ treatment (Fig. 2C) (see Supplementary Data 4). Inspecting Axis 2 of Fig. 2C, we found that the observed separation correlates with sequencing batch. Therefore, for any subsequent statistical modeling analysis (i.e., differential microbiota and gene expression modeling), sequencing batch information has been controlled for by including it as a fixed effect in the modeling statement.

To compare the effect of the two treatments to microbiome alpha diversity we calculated the Inverse Simpson Diversity

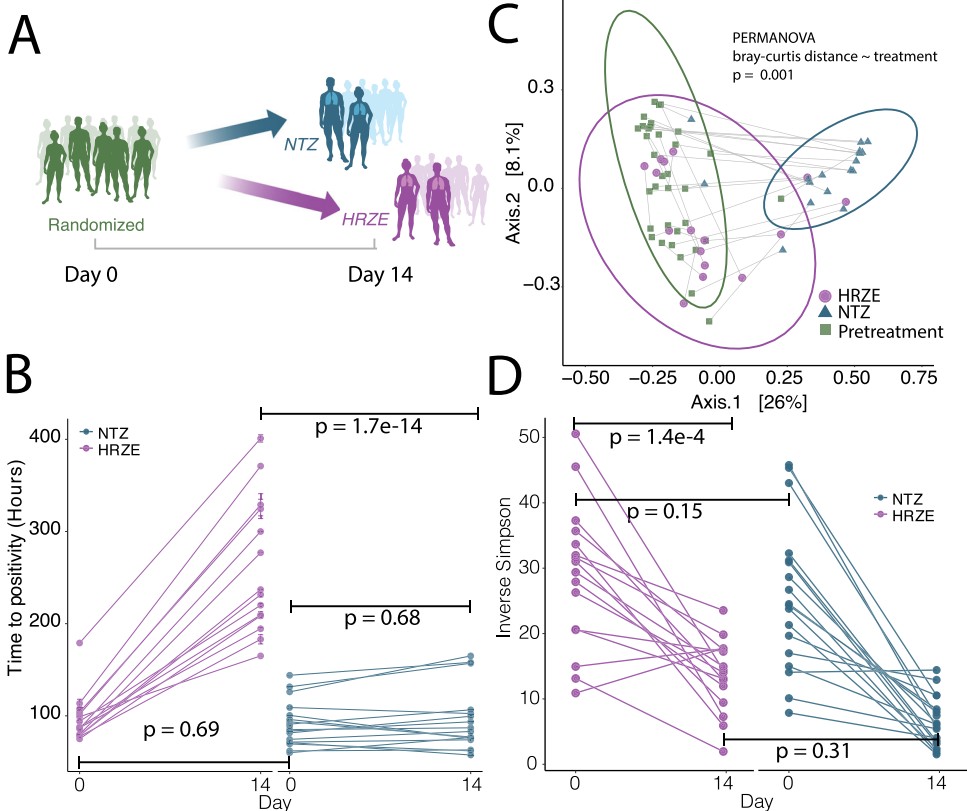

**Fig. 2 Both HRZE and NTZ perturb the gut microbiota after two weeks of therapy, but only HRZE reduces *M. tuberculosis* bacterial load. A** Schematic of the clinical trial comparing bactericidal effect of HRZE and NTZ. **B** Paired *M. tuberculosis* sputum time to positivity (TTP) at day 0 and day 14 for the NTZ treatment arm and HRZE treatment arm. Data are displayed as range (minimum and maximum) of two-three technical replicates; $n = 16$ biologically independent individuals for the HRZE arm and $n = 19$ biologically independent individuals for the NTZ arm. Linear mixed effect modeling was used to determine significance of difference in post/pre treatment in each arm as $TTP \sim 1 + Sex + Age + Treatment + Time + Treatment : Time + 1|ID$, where *Treatment* indicates the arm (NTZ or HRZE), *Time* indicates Pre or Post antibiotic administration, and : indicates the interaction term. For each individual we use as TTP measurement the average of the relative technical measurements. NTZ treatment is associated with no difference in TTP between day 0 and day 14 ($p > 0.05$ for the coefficient of variable *Time*, see Supplementary Data 2), whereas HRZE significantly reduces bacterial load ($p < 0.05$ for the coefficient of variable *Treatment : Time* see Supplementary Data 2). Data for TTP were obtained from Walsh et al.[26]. **C** Principal Coordinate analysis (PCoA) with Bray–Curtis distance showing differences in microbiome community structure between individuals before and after 14 days of either HRZE or NTZ treatment. The gray line connects baseline and day 14 treatment paired samples. PCoA1 clearly discriminates samples post NTZ treatment (pink triangles) from those at baseline or after HRZE treatment. PERMANOVA analysis was used to reject ($p = 0.001$, see Supplementary Data 4) the null hypothesis that the centroids and dispersion of the groups (pretreatment, after NTZ and after HRZE) are equivalent for all groups (see Supplementary Data 4). **D** Microbiota alpha diversity plotted using the Inverse Simpson Diversity index; $n = 16$ biologically independent individuals for the HRZE arm and $n = 19$ biologically independent individuals for the NTZ arm. Linear mixed effect modeling was used to determine the significance of difference of treatment on diversity. We fitted the model $Diversity \sim 1 + Sex + Age + Treatment : Time + 1|ID$. The symbol: indicates the interaction term. HRZE was used as the reference level. No significant difference between the two treatment at baseline was observed. Both groups ($p < 0.05$ for the coefficient of variable *Time* corresponding to HRZE treatment and $p < 0.05$ for the coefficient of variable *Treatment:Time* corresponding to NTZ treatment in this model) display significantly reduced Inverse Simpson diversity after 14 days of treatment (see Supplementary Data 3). Source data are provided as a Source Data file.

Index for every microbiome sample[28]. We then regressed the Inverse Simpson Diversity Index via linear mixed-effect modeling as $Diversity \sim Sex + Age + Batch + Treatment + Time + Treatment : Time + 1|ID$ (see "Methods" section). We found that there were no differences in alpha diversity at baseline between the two arms, while both treatments significantly reduced microbial diversity ($p < 0.01$, see Supplementary Data 3), with NTZ treatment not having a significantly different effect compared to HRZE ($p > 0.05$, for the interaction term, See Supplementary Data 3) (Fig. 1D).

**Taxonomic alterations in microbiome composition induced by antibiotics are more pronounced in NTZ-treated individuals.** To identify phylotypes significantly affected by each of the two treatments, we modeled the abundance of each sequencing

variant identified via dada2 (ASV) via linear mixed-effect modeling as $ASV_i(counts) \sim Sex + Batch + Group + 1|ID$ using Limma/Voom[29] (see "Methods"section). This model statement enables quantifying sex and sequencing batch-dependent effects in addition to establishing effects that are due to treatment group (pre-treatment, HRZE, NTZ). We used the $1|ID$ random effect to control for baseline differences among individuals. ASVs significantly affected by the treatments were determined using a Benjamini–Hochberg false discovery rate (FDR) adjusted $p$-value of 0.05 (see "Methods" section).

This analysis indicates that the HRZE effect on the gastrointestinal microbiota consisted of depletion of 82 ASVs, many belonging to genera from the Order Clostridiales (FDR < 0.05) (see Supplementary Data 5). Notably, many of these ASVs (e.g., *Blautia spp.*, *Butyrivibrio* spp., *Clostridium* spp., *Eubacterium* spp., *Faecalibacterium* spp., *Gracilibacter* spp., *Oscillibacter* spp.,

*Roseburia* spp., *Ruminococcus* spp., *Sporobacter* spp.) are known to be involved in a number of health-associated functions such as SCFA production[30] or bile acid transformation[31] (Fig. 3A, B). NTZ had a more substantial disruptive effect, indicated by depletion of 387 ASVs and enhancement of 16 ASVs. NTZ caused a reduction of a larger swath of Clostridiales, which included all but one of the ASVs depleted by HRZE, as well as many other additional Firmicutes, (FDR < 0.05). In addition, NTZ caused an expansion of pathobionts including, *K. pneumoniae, E. coli, C. freundii, S. alactolyticus*, and *E. faecium*[32] (FDR < 0.01) (Fig. 3A, B). These representatives of the Order *Bacilli* and of the family *Enterobacteriaceae* have been shown to be positively selected by antibiotic treatment as a result of higher antibiotic-induced Redox Potential, which includes an increased gut epithelial oxygenation[33–35]. For this reason, both in the literature and in this paper are referred to as oxygen-tolerant pathobionts (see[32]).

Taken together, these results demonstrate that NTZ, despite having no effect on *Mtb* bacterial load, causes a perturbation of the gastrointestinal microbiota which is more pronounced compared to that caused by HRZE. This includes the depletion of a large number of Clostridia, and the selection for known disease-associated, oxygen-tolerant pathobionts.

**HRZE and NTZ uniquely affect host peripheral gene expression.** As the above results highlighted a differential effect on TB disease resolution and gastrointestinal microbiota composition of the two antimicrobials tested, we recognized that this presented a unique opportunity to infer possible relationship between microbiome composition, *Mtb* bacterial load, and peripheral gene expression. As the patients were randomized at baseline before being assigned to the two treatments, we used linear mixed-effect modeling with Limma/Voom to model the abundance of each host transcript as $Gene_i(Counts) \sim Sex + Batch + Group + 1|ID$ (See "Methods" section).

We determined the functional pattern of treatment-induced changes in overall transcript abundance by performing gene set enrichment analysis (GSEA)[36,37] (See "Methods" section) on the ranked limma/voom expression data for the baseline vs post-NTZ or post-HRZE treatment contrasts. We used GSEA to estimate enrichment for the MiSigDB Hallmark pathways[36,38], which are intended to give a broad overview of biological pathways that may be expressed. In individuals undergoing HRZE treatment, we observed a significant (FDR < 0.05, see Supplementary Data 6) depletion at day 14 of inflammatory response, IFNα response, IFNγ response, TNFα signaling via NFκB, and IL6 JAK STAT3, all of which are consistent with the broad immunologic effects of antibiotic induced reduction in the levels of a bacterial pathogen[14,17,39] (Fig. 4A). In contrast, and to our surprise, NTZ treatment showed the opposite effect. Inflammatory signaling pathways reduced by HRZE, including TNFα signaling, IFNγ signaling, and type 1 interferon signaling were all significantly enriched by NTZ treatment at day 14 (Fig. 4A). Several other pathways such as hypoxia, apoptosis, and reactive oxygen species (ROS) that are considered hallmarks of immune dysregulation[40], were also enriched by NTZ treatment. As NTZ was found to perturb the microbiome while keeping *Mtb* bacterial load in the sputum (TTP) substantially unchanged, we hypothesized that the NTZ effect was likely a partial function of microbiome alteration (Supplementary Data 6).

To gain a deeper understanding of gene signatures affected by each of the two drugs, we first focused our analysis on a set of published transcriptomic markers of active TB from multiple human cohorts across sequencing platforms (microarray and RNAseq) that are differentially abundant between LTBI, active TB, and healthy control individuals[19]. In our study, we detected

363 of these 373 transcripts in pretreatment, active TB subjects. We defined three classes of changes to these transcripts with two weeks of HRZE or NTZ treatment: (1) *renormalization* (transcripts whose pre–post HRZE/NTZ fold change in expression displays the same sign, or direction, of the previously-reported fold-change between active TB and control/LTBI from ref. [19]); (2) *unchanged* (transcripts with no change in expression between pre-post HRZE/NTZ administration); and (3) *exacerbation* (genes whose pre–post HRZE/NTZ fold-change sign is opposite to the previously-reported fold-change between active TB and control/LTBI from ref. [19]). 157 of these active TB signature genes were significantly affected by HRZE (FDR < 0.05) (Fig. 4B, Supplementary Data 7). Of these 157 affected genes, 144 (92%) were found to renormalize with the treatment (i.e., displaying the same direction of the fold change reported for active TB vs. control individuals), while 13 (8%) were found to exacerbate (i.e., opposite direction of the fold change). On the other hand, only four of these TB-related inflammatory genes were found to be affected by NTZ, and all of them (100%) were in the exacerbation category (Fig. 4C, Supplementary Data 8).

Because NTZ perturbs the microbiome without affecting *Mtb* disease resolution, we hypothesized that there may be other subsets of host transcripts that are linked to microbiome-dependent immunity that could be responsive to the observed microbiome perturbations. To test this, considering the already well-established link between microbiome dysbiosis and autoinflammatory conditions such as Inflammatory Bowel Disease (IBD)[42], we selected a recently published panel of 880 genes differentially expressed in colon biopsies from IBD patients compared to asymptomatic controls[41]. Of these 880 genes, 364 were detected in our dataset (expected given that we are profiling whole blood transcriptomics, rather than colon tissue). We defined renormalization as those genes that have a post/pre fold change due to antimicrobial treatment of the same sign as control/active IBD. Despite the more limited effect on the microbiome compared to NTZ, HRZE administration was found to be responsible for a change in 117 of these genes (FDR < 0.05) (Fig. 4D, Supplementary Data 9), while NTZ was found to be responsible for a change in 55 of them (FDR < 0.05) (Fig. 4E, Supplementary Data 10). These data suggest that there may be microbiome related changes in the peripheral inflammatory state that are induced by HRZE and NTZ.

**Microbiota and peripheral inflammatory profiling of a longitudinal, observational HRZE treatment cohort.** To validate the observations obtained after two weeks of HRZE and to identify effects that may occur with increased treatment length, we enrolled a longitudinal treatment cohort (20 individuals) to measure disease resolution over the course of the 6 months of HRZE treatment, with periodic sampling of sputum for TTP, stool for microbiome profiling, and peripheral blood for RNA sequencing (Fig. 5A). All participants received standard of care HRZE therapy (isoniazid 300 mg daily, rifampin 600 mg daily, pyrazinamide 25 mg/kg daily, and ethambutol 15 mg/kg daily). Sputum mycobacterial load (TTP) was collected at baseline, day 7, day 14, 1 month, 2 months, and 6 months at the completion of treatment. Stool from microbiome profiling was collected at each of these timepoints as well. Whole blood was collected at baseline, day 14, and 2 months. Stool and peripheral blood samples were processed for microbiome and transcriptomic profiling as for the clinical trial described above (See "Methods" section).

Similar to our observations in the HRZE arm of the HRZE/NTZ trial, TTP was significantly higher compared to baseline after two weeks of treatment (*p* < 0.05). TTP also significantly increases after two months compared to the two-week timepoint,

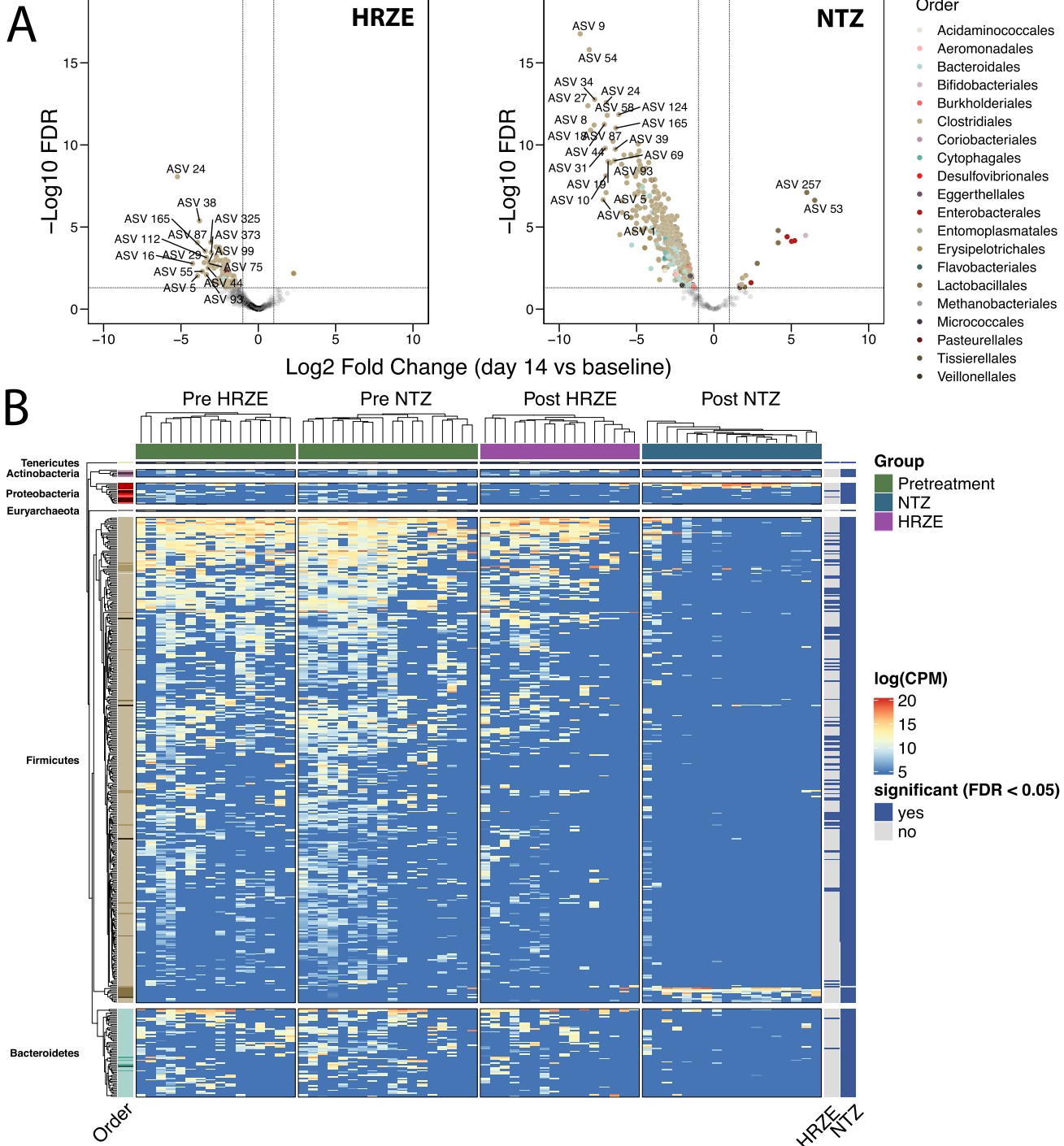

**Fig. 3 Overlapping and distinct microbiome perturbation induced by NTZ and HRZE. A** Volcano plots indicating the post (day 14) vs pretreatment (baseline) differences at the ASV level for HRZE and NTZ. The color of each ASV is according to the phylogenetic Order. A single linear mixed effect model for each ASV of the form $ASV_i(counts) \sim Sex + Batch + Group + 1|ID$ was fitted to determine differences due to treatment while accounting for sequencing batch and sex. ASVs significantly affected by the treatment were those determined to have a Benjamini-Hochberg false discovery rate (FDR) adjusted p-value less than 0.05 for the variable *Group* in the limma/voom model (see "Methods" section). The horizontal dotted lines indicate FDR < 0.05 and vertical dotted lines indicate |log2FC| > 1.5. **B** Within-arm unsupervised hierarchical clustering of the abundances of 404 ASVs found to be significantly affected by HRZE or NTZ treatment (FDR < 0.05, see Supplementary 5). The heatmap columns are split by arm membership (including baseline randomization group), and the heatmap rows are split by ASV phylogenetic Phylum, and within the Phylum, the Order is colored as in **A**. The right annotations (HRZE and NTZ) indicate whether each ASV was significantly perturbed by treatment. *P* value in *y* axis is adjusted according to Benjamini–Hochberg (FDR). Source data are provided as a Source Data file.

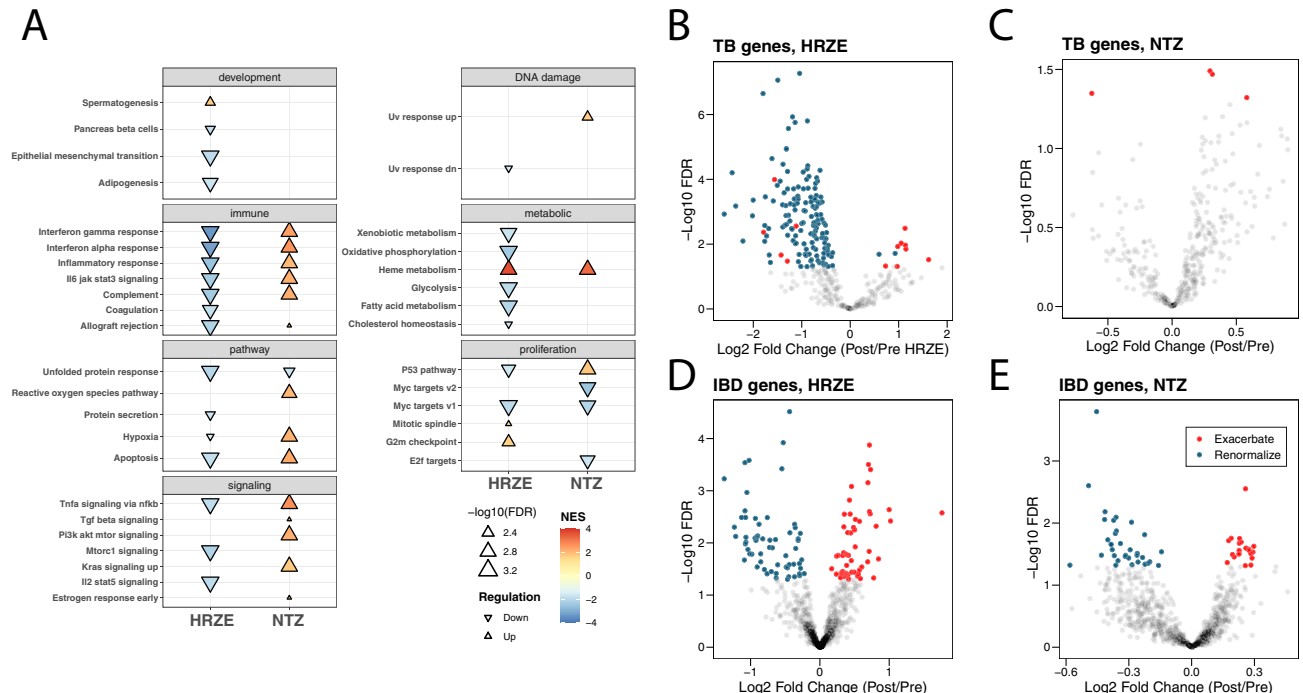

**Fig. 4 Hallmark pathway gene set enrichment analysis and gene expression comparison in HRZE and NTZ treated arms. A** Hallmark gene pathway changes associated with 2 weeks of HRZE (**A**) or NTZ (**B**). Positive are pathways overrepresented at 2 weeks of therapy (up), and negative are pathways underrepresented at 2 weeks (down), both compared to baseline. All pathways are significant (FDR < 0.05, see Supplementary Data 6) with the size of the arrow indicating level of significance. Only pathways from MiSigDB Hallmark pathway set found to be significantly altered in this analysis are shown. Here NES stands for Normalized Enrichment Score. **B**, **C** TB-associated peripheral blood transcripts from ref. [19], highlighting post treatment vs baseline changes in gene expression for HRZE (**B**) or NTZ (**C**). Notably, HRZE renormalizes (i.e., towards a healthy control state) the expression of 144 validated TB inflammatory transcripts and exacerbates only 13 (see Supplementary Data 7). NTZ is found only to exacerbate four (see Supplementary Data 8). TB-associated peripheral blood transcripts significantly affected by each treatment were those determined to have a Benjamini–Hochberg false discovery rate (FDR) adjusted $p$-value less than 0.05 for the variable *Group* in the limma/voom model (see "Methods" section). **D**, **E** Effect of HRZE and NTZ on blood gene expression for a set of IBD-associated genes from Palmer, et al.[41]. Both HRZE (**D**) and NTZ (**E**) cause different genes to either renormalize (HRZE 66, NTZ 34) (see Supplementary Data 9) or exacerbate (HRZE 55, NTZ 21) (see Supplementary Data 10). IBD-associated peripheral blood transcripts significantly affected by each treatment were those determined to have a Benjamini–Hochberg false discovery rate (FDR) adjusted $p$-value less than 0.05 for the variable *Group* in the limma/voom model (see "Methods" section) $P$ value in $y$ axis is adjusted according to Benjamini–Hochberg (FDR). Source data are provided as a Source Data file.

confirming the expected additional sterilization effect of antibiotics on *Mtb* ($p < 0.05$, linear mixed-effect modeling, see "Methods" section) (Fig. 5B, see Supplementary Data 11). Longitudinal analysis of microbiome composition revealed that diversity drops after just 7 days of treatment, increases after 1 month of treatment but remains significantly lower compared to baseline at the 6-month follow-up timepoint ($p < 0.05$, linear mixed-effect modeling, see "Methods" section) (Fig. 5C, see Supplementary Data 12).

We performed Limma/Voom differential analysis to determine effects of treatment and time on the microbiome and peripheral host transcriptomics using the same approach as above. As observed above and in our previous work[22], HRZE was found to depress Clostridiales after one week of treatment, with most of these Clostridiales ASVs remaining significantly depressed compared to baseline, even at the 6-month follow-up time point (Fig. 5D, see Supplementary Data 13). Overall, at day 7 compared to baseline, 19 ASVs were depleted and 3 were increased in abundance, at day 14 compared to baseline, 61 ASVs were depleted and 2 were increased in abundance, at one month compared to baseline, 83 ASVs were depleted and none were increased in abundance. Thus, during the first month of treatment, microbiome depletion relative to individual baseline samples was evident, however, later in the course of treatment, we observed a different trajectory. Relative to baseline,

compared to day 0 at two months of HRZE, we observed 11 ASVs depleted in abundance, while 53 ASVs increased in abundance. At the 6-month mark of HRZE treatment, compared to day 0 we observed 3 ASVs depleted in abundance while 327 ASVs increased in abundance (Fig. 5D, see Supplementary Data 13). The increased abundance of specific ASVs at the two month and 6-month mark appeared to be relatively heterogeneous between individuals. Importantly, only 7 and 2 Clostridia ASVs that were depleted in the first month of treatment did not recover at 2 and 6 months, respectively, with all other returning at a level not significantly different from day 0. In addition, a number of Clostridia ASVs which were undetected at day 0 appears to be enriched at the two later time points. Overall, this longitudinal analysis suggests that after one month of HRZE therapy, most ASVs that would be affected by therapy are depleted but also recover between two and six months of treatment (see Supplementary Data 13). In addition, it shows that there may be a replacement in community membership by phylogenetically-related ASVs that were undetected at day 0 (see Supplementary Data 13).

With respect to the peripheral host inflammatory profiling, we observed distinct changes in gene signatures at two weeks (day 14) and two months of treatment, compared to baseline (see Supplementary Data 14). We observed a similar decrease in common inflammatory pathways in the Hallmark pathway

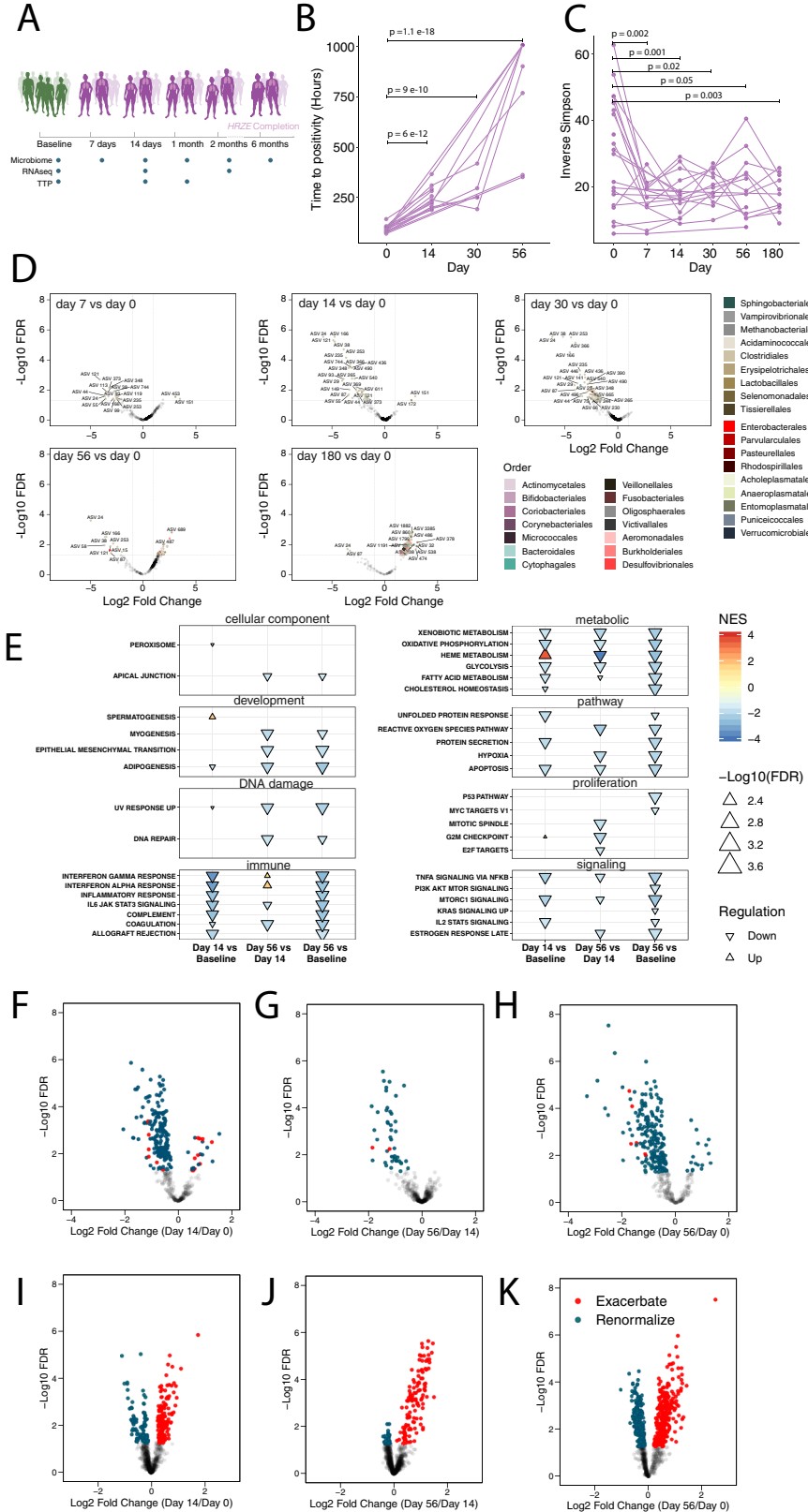

dataset described in the HRZE/NTZ trial (Fig. 5E, see Supplementary Data 15). Interestingly, comparing Day 56 to baseline or to Day 14, we see additional reduction in inflammatory gene signatures, potentially explained in part by the further increase in TTP (reduction of bacterial load) at Day 56 (Fig. 5F–K, see Supplementary Data 16, 17).

*Multi-omics-constrained mathematical modeling to decouple the contribution to peripheral inflammatory signature of gastro-intestinal microbiota and* Mtb. We next sought to determine the relative contribution of gastrointestinal microbiota and *Mtb* dynamics in predicting the changing dynamics of peripheral inflammatory gene signature. As a granular reflection of the TB

**Fig. 5 Longitudinal profiling of HRZE treatment induced changes of microbiome composition and peripheral gene inflammatory expression. A**. Schematic diagram of Cohort. **B**. Time to positivity was measured at baseline ($n = 19$ biologically independent samples), day 14 ($n = 14$ biologically independent samples), one month ($n = 5$ biologically independent samples), and two months ($n = 9$ biologically independent samples). To determine statistical significance of differences in TTP at different time points we fit the linear mixed-effect model $TTP \sim Sex + Age + Time + 1|ID$. We inspected the $p$-value associated by running contrasts for the variable *Time* to determine significant ($p$-value for the contrast <0.05) differences in TTP. **C** Microbiome diversity was computed for each study volunteer at baseline ($n = 20$ biologically independent samples), day 7 ($n = 7$ biologically independent samples), day 14 ($n = 19$ biologically independent samples), one month ($n = 13$ biologically independent samples), two months ($n = 13$ biologically independent samples), and 6 months ($n = 13$ biologically independent samples). Microbiome α diversity was measured using the Inverse Simpson index[43]. To determine statistical significance of differences in α diversity at different time points we fit the linear mixed-effect model $Diversity \sim Sex + Age + Time + 1|ID$. We inspected the $p$-value associated by running contrasts for the variable *Time* to determine significant ($p$-value for the contrast <0.05) differences in microbiota diversity **D** Volcano plots showing significance of differences in microbiome composition vs. fold change from baseline at Day 7, Day 14, Day 30, Day 56, and Day 180. **E** Normalized enrichment scores calculated for the Hallmark Pathway list for Day 14 vs. Baseline, Day 56 vs Day 14, and Day 56 vs. Baseline. **F–H** Volcano plot showing TB transcripts from ref. [19] at Day 14 vs. Baseline at Day 56 vs. Day 14, and at Day 14 vs. Baseline. **I–K** Volcano plot showing transcripts from Palmer et al.[41] of IBD cases vs. controls detected in this study for Day 14 vs. Baseline, Day 56 vs. Day 14, and Day 56 vs. Baseline. Source data are provided as a Source Data file.

inflammatory disease state, we reasoned that the resolution of this state would likely be influenced by the combined upstream factors of pathogen killing and microbiome perturbation. We hypothesized that modeling these relationships could be achieved by both leveraging individual variability in response to HRZE treatments, as well as by leveraging the data from the NTZ arm of the HRZE/NTZ trial. This is because the NTZ arm is effectively a "natural" experimental control as it provides for examples in where we observe both gastrointestinal microbiota and gene-expression changes in the absence of changes in *Mtb*. Specifically, our aim was to identify microbiota members for which changes in abundance associates with significant changes in inflammatory pathways in the three groups of antibiotic-treated individuals. To do this, for each inflammatory hallmark pathway identified to be significantly affected by HRZE or NTZ via linear mixed effects modeling, we first computed the change in normalized enrichment score between two consecutive time points $t_\psi$ and $t_{\psi+1}$ for individual $s$ as $\frac{y_{lt_{\psi+1}s} - y_{lt_\psi s}}{t_{\psi+1} - t_\psi}\Big]_{l=1\ldots L}$. We then regressed this quantity against the corresponding fold change in abundance for every ASV $v$ in the same interval $\frac{x_{vt_{\psi+1}s}}{x_{vt_\psi s}}\Big]_{v=1\ldots N}$ and against the corresponding fold change in TTP, $\frac{p_{t_{\psi+1}s}}{p_{t_\psi s}}$. Using change from baseline values accounts for the random effect of each subject without having to incorporate this into the model statement. We solved this regression problem using Random Forest regression as in ref. [44]. To train the models, we used all the observations from the HRZE/NTZ trial and from the longitudinal observational cohort for a total of 34 paired samples (Fig. 1). We fit a model for each inflammatory pathway using all the data from the two longitudinal cohorts (HRZE/NTZ trial, and HRZE longitudinal observational cohort) because we wanted to find patterns that are general across multiple datasets. Each model was trained using 5000 trees and with a train-validation partitioning of 80–20% of the data. We reasoned that this approach was appropriately suited for this type of "large p, small n" multi-omics dataset common in clinical research[45]. Importantly RFR modeling has significant advantages compared to traditional multi-linear regression techniques, because it is agnostic to model structure (e.g., non-parametric regression), it does not need to meet common assumptions underlying classical regression techniques, and is able to intrinsically perform ranked feature selection. Importantly, while the interpretation of RFR is apparently less immediate compared to traditional regression (e.g., there are per se no regression coefficients or betas), downstream analysis, which includes Permutated Importance[46] and Accumulated Local

Effects (ALE) calculations[47] (see "Methods" section) allows for the estimation of the significance of predictors (e.g., TTP, microbiome constituents, etc.) and of their effects on the dependent variable (e.g., host peripheral inflammatory markers).

When plotting the average slope of the ALE curves for predictors with significant Permutated Importance values ($p < 0.05$, see Supplementary Data 18), our analysis identifies the increase in TTP (and therefore a decrease of *Mtb* in the sputum) and the increase in abundance of ASVs from Clostridia, especially members from the Cluster IV and XVIa groups[48], which have been shown to induce anti-inflammatory responses (e.g., Treg-induction)[2,3] through SCFA production[49] including *E rectale*, *F. prausnitzii*, *G. formicilis*, *E hallii*, *O ruminantium*, *D. formicerans*, *S. variable*, *B. faecis*, and *B. obeum*. The abundance of these microbiome components was associated with reduction in peripheral proinflammatory response including INFγ, INFα, Inflammatory Response, IL6 JAK STAT3 Signaling (Fig. 6). In contrast, increased abundances of *E. coli* and *E. faecium* were associated with inflammatory exacerbation (Fig. 6), consistent with a large body of literature demonstrating that gastrointestinal overgrowth of these species is the hallmark of gastrointestinal dysbiosis and inflammation[50,51] and often associates with adverse clinical outcome[52,53].

Taken together, our data and related computational analyses show that the changes in inflammatory gene expression that accompany treatment of TB correlate with the anti-microbial activity of the drugs that lead to pathogen clearance and with antibiotic-induced changes in microbiome composition. Based on this modeling result, we propose two modules of microbiome-inflammatory effects. The first would be the exacerbation of TB-associated inflammation by depletion of Clostridia (especially Cluster IV and XIVa), which is evident in both the HRZE and NTZ groups. In addition, the enhancement of pathobionts such as *E. faecium*, *S. alactolyticus*, and *E. coli*, which only occurs with NTZ, may also exacerbate inflammatory pathway expression within an individual. Importantly, as our model explicitly accounts for the abundance of all treatment-affected ASVs (i.e., we are controlling for the presence and abundance of all bacteria by including them in the model as predictors), the fact that these pathobionts' abundance correlates with inflammatory response exacerbation suggests that the identified microbiome-immune associations may not solely reflect Clostridia dynamics.

Based on this modeling, we hypothesize that successful disease resolution may be associated with preservation of Clostridia, whereas their depletion and consequent enhancement of dysbiosis-associated *Enterobacteriaceae* or Bacilli pathobionts might slow resolution or even support inflammatory exacerbation[54].

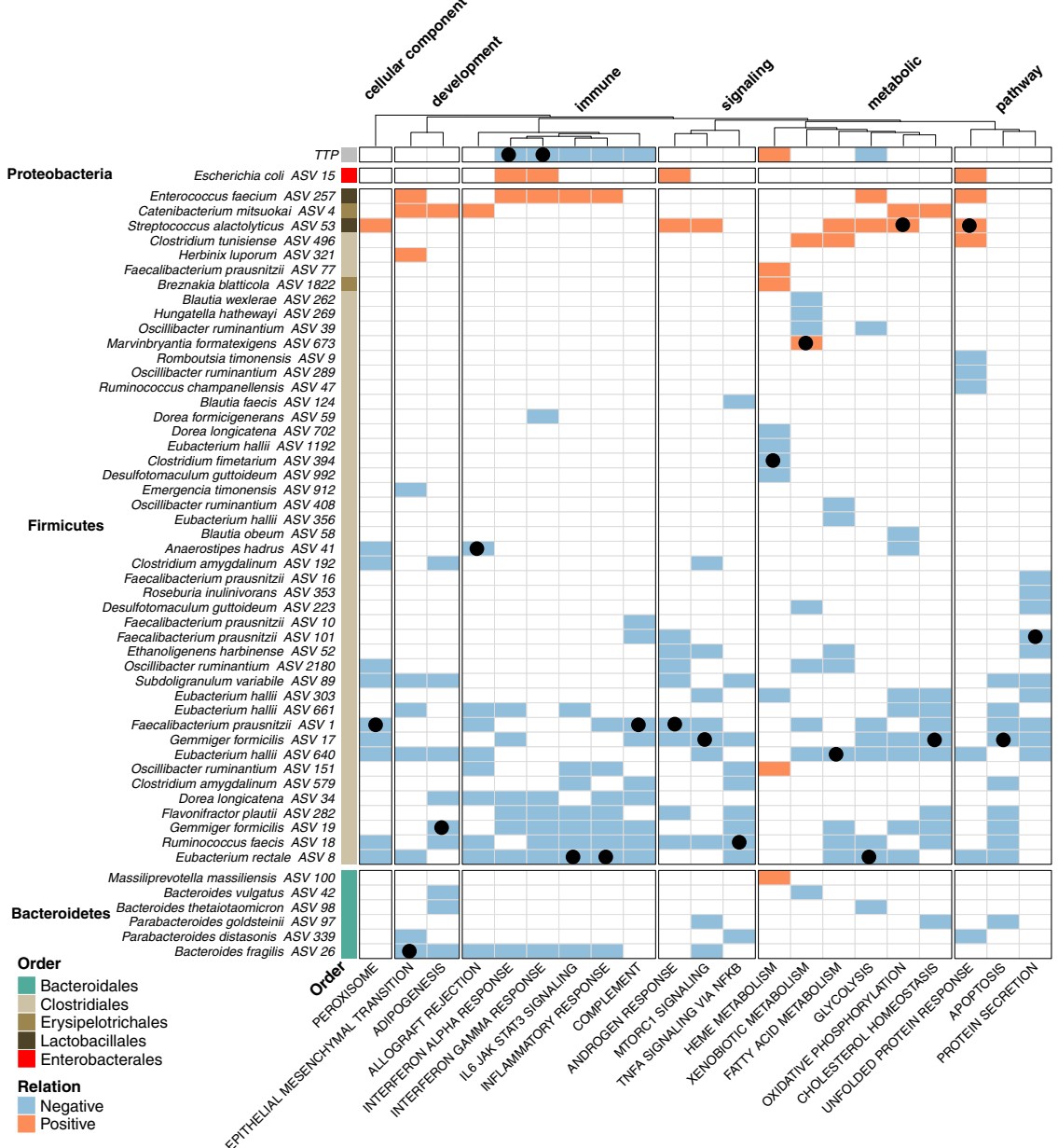

**Fig. 6 Use of Random Forest Regression Modeling to search for associations between immune-related peripheral blood gene signatures and changes in gastrointestinal microbiota and TTP.** The heatmap displays the sign of the derivative of the ALE curve (See Text). Blue/orange entries indicate features found to significantly associate with changes in a specific inflammatory pathway. Blue indicates a negative relationship, while orange a positive. For each immune pathway, a pathway-ASV association was determined significant if the Benjamini–Hochberg false discovery rate (FDR) adjusted p-value from the permutated importance analysis was found to be less than 0.05. Black symbols are used to identify the modeling-identified top important predictor (i.e., the predictor that if missing would lead to highest increase in mean squared error between model predictions and observations) for each specific host pathway. This analysis shows that reduction in TB burden and increased abundance of health-associated Cluster IV and XIVa Clostridia predicts inflammatory dampening. In contrast, increased abundance of oxygen-tolerant pathobionts including *Enterococcus*, *Streptococcus*, and *E. coli* predicts inflammatory exacerbation. A table reporting variable importance values, slope and intercepts from the ALE plot calculation and the related p values is provided in Supplementary Data 18. Source data are provided as a Source Data file.

**Relationship of the microbiome and peripheral gene expression in a healthy control validation cohort.** The results from our machine learning modeling on the data from both longitudinal treatment cohorts provide support to the hypothesis that specific gastrointestinal microbiota members are associated with immune-related peripheral blood gene signatures in humans. Specifically, higher abundance of Clostridia is negatively associated with inflammation (e.g., INFα, INFγ, IL6/JAK/STAT3, Inflammatory Response gene signatures), while higher abundance of common

oxygen-tolerant pathobionts is associated with exacerbation of these signatures. To assess the generality of these findings we hypothesized that, even in healthy individuals, different levels of colonization by these bacteria would associated with different levels of immune-related peripheral blood gene signatures.

To test this hypothesis and to ultimately validate the finding from the modeling, we analyzed a set of human data from two healthy control cohorts. A subset of these data was reported in previous work from us[5], and come from a cross-sectional study of

TB negative healthy household contacts of active pulmonary TB patients (termed Family Contacts, FC) and healthy unexposed donors from the same community in Haiti (termed Community Controls, CC) (see "Methods" section). For these two cohorts we have a total of 55 healthy control individuals (19 FC and 36 CC) for which we gathered both microbiome 16S rRNA sequencing data and peripheral blood transcriptomics.

We first wanted to validate our findings that peripheral blood transcriptomic patterns that renormalize after HRZE treatment and exacerbate after NTZ treatment (which were determined from our comparison to published Control/Active TB gene signatures datasets) remain valid when comparing to peripheral inflammatory profiling in our validation FC and CC control cohorts (Cohort 3). To link transcript abundance to immune pathway enrichment we utilized single-sample gene set enrichment analysis with the GSVA package in R (see "Methods" section)[55]. We first performed unsupervised clustering on the samples-by-pathway NES scores for all samples in this study and found that individuals from different cohorts have broad qualitative differences in distinct biological pathways (Fig. 7A). We then regressed the Euclidean distance between every pair of FC/CC and pre-treatment or during treatment samples against Time, Treatment and patient ID using linear mixed effect modeling for the HRZE/NTZ trial and the long-term observational cohort independently (see "Methods" section). We found that in both the HRZE/NTZ trial and the long-term observational cohort, HRZE significantly reduced the distance to FC/CC samples ($p$ value for HRZE in HRZE/NTZ trial $<1e-20$, $p$ value for Day 14 longitudinal, observational HRZE treatment cohort $<1e-10$, $p$ value for Day 56 longitudinal, observational HRZE treatment cohort $<0.05$, Supplementary Data 19 and Supplementary Data 20) compared to pre-treatment (Day 0), confirming a peripheral inflammatory renormalization trend. In contrast, NTZ was found to increase the distance to FC/CC compared to pre-treatment $p$ value for NTZ in HRZE/NTZ trial $< 1e-10$, see Supplementary Data 19), again validating an NTZ-induced inflammatory exacerbation.

We performed RFR (see "Methods" section) for each pathway against the microbiome space, and summarized the findings in Fig. 7B. Surprisingly, we found the abundance of a large number of Firmicutes and particularly Clostridia to be associated with a number of the characterized Hallmark molecular pathways (see Supplementary Data 21). Even more intriguingly we found that higher abundance of ASVs that are mapped to health-associated Clostridia including *F. prausnitzii*, *Rumonoccocus* spp., and *C. catus*, are associated with a reduction in proinflammatory pathways including INFα, and INFγ. This independent analysis in homeostatic conditions is consistent with the findings obtained from the application of our modeling analysis on the HRZE/NTZ clinical trial and on the longitudinal, observational HRZE treatment cohort (Fig. 6). These results, performed in a cohort of 55 volunteers from the Haitian community, thus reinforces the hypothesis that the relative abundance of specific gastrointestinal microbiota members, which we find to be perturbed by TB therapy, correlates with inflammatory peripheral gene expression in humans.

## Discussion

Since the advent of high-throughput microbiome characterization, it has become clear that antibiotics are one of the most common and severe perturbing influences on human microbiome composition, with both acute and longer lasting effects[56,57]. It also has become evident that the specific microbiome constituents have specific effects on host immunity, including influencing the abundance and function of immune cell subsets[24]. Significant

prior data have documented the effects of antibiotics on microbiome composition and function and the consequent influence of these microbiome factors on immune cell populations[58], with the majority of these findings derived using in vivo mouse models. While there is no doubt that microbiota dynamics affect host immunity[6], it remains unknown to what degree antibiotic induced perturbation of the microbiome may modify the outcome of treatment of infection, or what relationships exist in humans between gut microbiota composition and peripheral gene expression. It is conceivable that antibiotics work to clear infection both due to direct pathogen killing and by immune modulation through the microbiome. It is also possible that the pathogen killing effect of antibiotics may be partially counteracted by detrimental immune effects induced by microbiome perturbation. Such dynamics may be particularly relevant to the treatment of chronic infections such at tuberculosis, in which antibiotic therapy is prolonged and the disease manifestations reflect a mixture of pathogen burden and the balance of inflammatory mediators that cause tissue destruction and chronic symptomatology[59,60].

Antibiotic-sensitive tuberculosis is treated with six months of antibiotics with predominantly mycobacterial specific agents. In this study we report the early and late microbiome effects of HRZE therapy in subjects with active TB and demonstrate that the same changes observed in a human cross-sectional study of TB treatment[22] (comparing vs. cured and LTBI individuals) were present after just two weeks of treatment. As previously shown[22], we conclude that HRZE treatment has a rapid and narrow effect on the gastrointestinal Class of Clostridia, a finding that was also demonstrated in mice[21,61]. We note that given the mycobacterial-specific nature of TB drugs, and the combinatorial nature in which small molecules interact to affect the microbiome, it was difficult to predict that primarily Clostridia, in the Phylum Firmicutes, would be targeted by HRZE therapy, whereas Actinobacteria, the phylum to which *Mtb* belongs, are relatively unaffected. Experiments in mice have demonstrated that this anti-Clostridia effect is primarily driven by rifampicin/PZA[21]. Clostridia are immunologically active components of the microbiota through their production of metabolites such as short chain fatty acids and other compounds[2,5,6,62,63].

The NTZ/HRZE study allowed us to dissect the relative contributions of pathogen killing and microbiome perturbation to disease resolution because one treatment arm, standard therapy, both reduced *Mtb* bacterial burden and perturbed the microbiome, whereas NTZ had no effect on average *Mtb* burden, but did perturb the microbiome in a fashion that overlapped with HRZE. We extended these findings in the longitudinal dataset which followed subjects with active TB for one week, two weeks, one month, two months, and 6 months post treatment. These analyses may provide support to the hypothesis that antibiotic perturbation of the microbiome has systemic effects on peripheral gene expression. Further, we speculate that the large heterogeneity in the rapidity of treatment response in TB may be a partial function of heterogeneity in the effects of antibiotics on the microbiome.

To validate the inferred microbiome-host inflammatory relationship, we mined microbiome and blood transcriptomic profiling from an independent human cohort of healthy Haitian individuals. Remarkably, despite the reduced peripheral levels of inflammatory pathways compared to subjects with active TB, we again observe that higher abundance of members of Clostridium IV and XIVa is associated with a reduction in the expression in pro-inflammatory pathways. This result supports our conclusion that microbiome composition sets the tone of systemic inflammation, both in disease states and in homeostatic conditions. Further, it is consistent with the prior findings in both humans

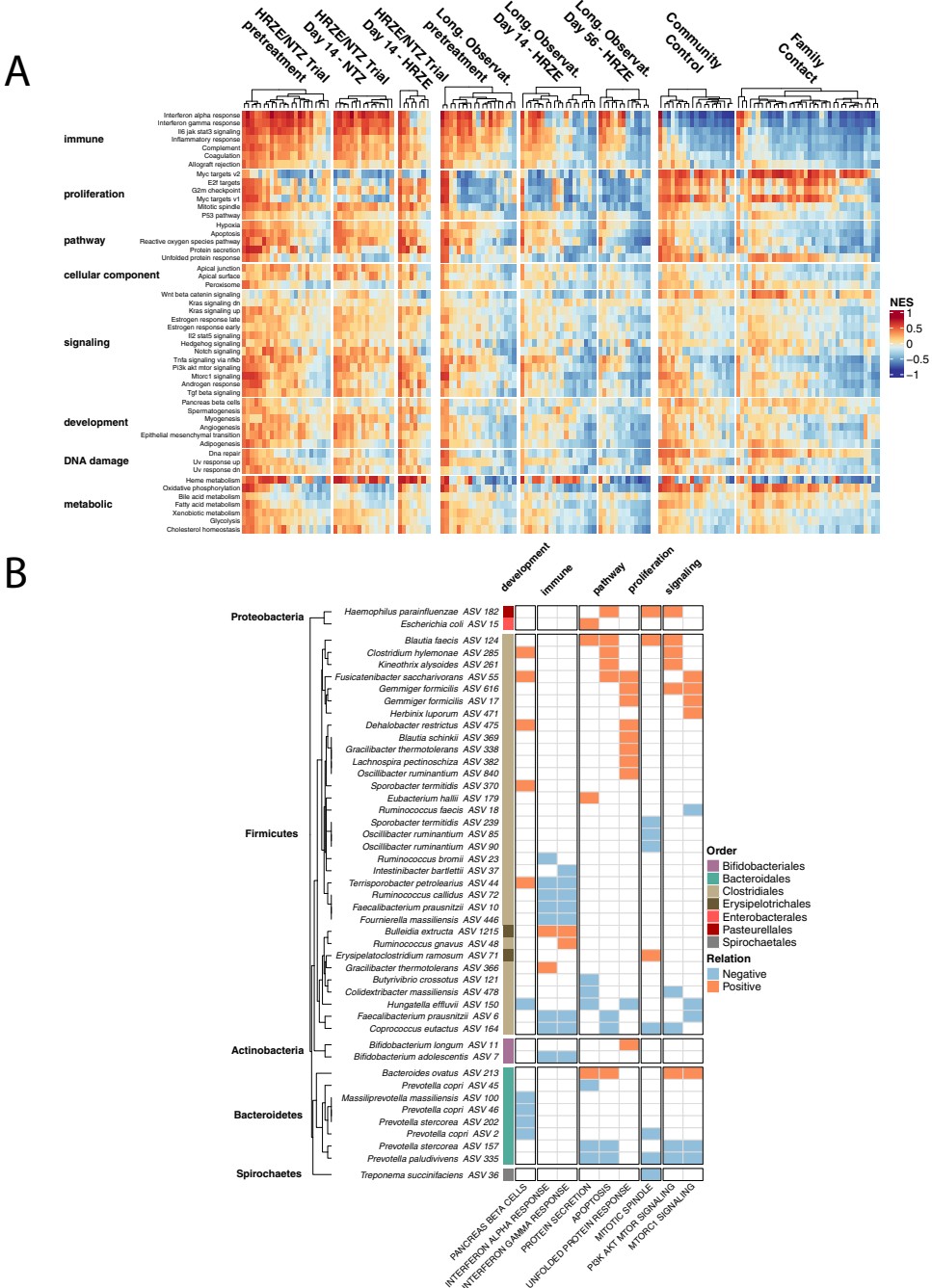

**Fig. 7 Analysis of microbiome and blood peripheral gene expression in an independent healthy control human cohort validates association between specific microbiome members and host peripheral gene expression. A** NES scores of 50 Hallmark pathways from the MiSigDB on a per-sample basis for all cohorts in this study. NES score was calculated using the variance stabilized transformed counts from DSEeq, calculated with the GSVA package in R, and plotted after scaling (Z score) across all samples. Columns are split based on arm or group membership and rows are split based on Hallmark pathway categorization. **B** Random forest regression results associating specific microbial taxa with Hallmark pathways. Only pathways identified in the RFR model are shown. The 'Relation', calculated by taking the first derivative of the ALE plot for each relationship, is positive if the pathway positively associates with a particular ASV, or negative if the pathway negatively associates with a particular ASV. Source data are provided as a Source Data file.

and animals that Clostridia have been associated with induction of anti-inflammatory states[64,65].

Finally, given the challenge of explaining the relationship between microbiome composition and peripheral gene expression with paired samples, randomized to drug combinations with vastly different effects on both body systems, we strove to use appropriate mathematical approaches for this type of analysis.

For single-omic microbiome and RNAseq data, we chose to use limma/voom to model changes in these data given their ability to model many effects while using subject's baseline values as random effects. For multi-omics integration, we used Random Forest Regression. While there are a variety of statistical and machine learning techniques able to investigate relationships between complex multiparametric datasets ("large p": microbiome

composition, RNAseq data, clinical metadata, randomization cohort, paired-sample baseline normalization, etc., and a "small n" of individuals in early phase clinical trials), Random Forests are adequate for microbiome purposes, as they have been shown to outperform Support Vector Machines in some instances, especially for continuous variable data, and need initialization of a smaller set of parameters compared to other deep-learning methods. We believe that our results highlight the utility of these models in two ways: (1) By providing evidence for or against a particular hypothesis about clinically significant relationships between many potentially related parameters, and (2) By providing hypothesis-generating relationships between the multi-omic constituents (i.e., features) of these models, which can be further tested in mice, validation cohorts, or other model systems.

One of the major goals of this study was to evaluate our ability to infer associations between microbiome, pathogen, and host inflammatory state in humans by using measurements that are rapid and not invasive (i.e., stool DNA sequencing, sputum bacterial load, and blood RNA sequencing). The underlying rationale is that identification of these associations could lead to follow-up mechanistic work both in vitro and in vivo aimed not only at validating them but also at prototyping microbiome intervention strategies that could improve inflammatory resolution during TB treatment. By doing so however, we made the implicit assumption that TTP is a sufficient predictor variable to account for the effect that the disease has on the systemic immune response. Another caveat to consider is our lack of quantitation of cellular immunity. While this information can be obtained from in vivo experiments which, for example, have shown that the impact of microbiome dysbiosis from TB treatment is at the level of alveolar macrophages, thus suggesting that gut microbiome alterations may have not only impact systemic inflammatory responses but also affect lung-resident immune cells, it is something that is at the moment not doable in the types of field settings on which we are basing our analysis. We therefore acknowledge that TTP quantitation as a window in the disease state is not a perfect parameter, that our results need to consider this as a possible caveat and point to the fact that future work may be geared to consider developing non-invasive and rapid assays for other correlates that could provide additional levels of information on host disease state. In addition, we acknowledge that any associations observed in this study may not account for latent variables or confounding factors that may explain microbiota-inflammatory-pathogen relationships. Finally, we acknowledge that, despite being performed on an independent cohort, our analysis to validate the identified microbiome-peripheral gene expression associations was applied on a healthy control cohort. Validation of signatures on people who do not have the target condition but suffer from other diseases may be expected to produce false-positive results more often than otherwise healthy individuals[66]. As such, future work will be geared to determine presence of the identified relationships in new independent TB treatment cohorts.

Despite the above caveats, our data indicates that within the first 14 days of treatment of TB, resolution of the active inflammatory response of TB (as measured by peripheral blood transcriptomics) may be affected both by reducing Mtb burden, as well as through antimicrobial-induced microbiome perturbations that may act directly on systemic immune function. Among the pathways tightly correlated with both factors are the signature activated pathways of active TB disease: IFNγ, type I interferon, and TNFα[14]. There is growing evidence that the outcome of active TB reflects a mixture of pathogen burden and cytokine networks that include IL-1 and IFNγ, with the latter acting to exacerbate disease[59]. Our findings indicate that the microbiome perturbation that accompanies TB treatment is a predictor of the

normalization of these same pathways during early treatment, suggesting that microbiome perturbation could modify or predict the rapidity of disease resolution. In the first two weeks of treatment, pathogen killing is the dominant factor, but microbiome-dependent modulation of inflammatory responses during treatment may assume an important role during the later phases of treatment when pathogen killing slows. The validation of the relationships between microbiome composition and peripheral gene expression in a healthy control cohort, especially for the collective expression of these same pro-inflammatory and anti-inflammatory pathways, suggests that these relationships may extend into other populations. Whether these relationships are causal, or biomarkers of another state, will remain at the forefront of future study design. Future work will be directed to applying the analytical tools and study design presented here to later time points in the TB treatment course to examine whether microbiome perturbation during treatment associates with clinically relevant treatment outcomes, and whether the abundance of Clostridia correlates with rapidity of Mtb sterilization or the resolution of the inflammatory response that accompanies active TB. Such data might help support trials to test microbiome monitoring as a predictor of TB treatment outcome or help understand interindividual heterogeneity in treatment outcomes.

## Methods

**Ethical statement and study approval.** All volunteers provided written informed consent to participate in this study. All human studies were reviewed and approved by the IRBs of both Weill Cornell Medicine and Groupe Haitien d'etude du Sarcome de Kaposi et des Infections Opportunistes (GHESKIO) Centers (Port-au-Prince, Haiti). Participants provided informed consent prior to peripheral blood draw for whole blood collection and stool collection for 16S rDNA sequencing. All methods and procedures were performed in accordance with the relevant institutional guidelines and regulations. Supplementary Data 1 provides a table with first-last day of enrollment for each Cohort.

### Donor recruitment and protection of human subjects

*Secondary analysis of published HRZE/NTZ Clinical trial NCT02684240.* All randomization procedures, statistical plan including power calculation and all other details of the NCT02684240 trial are reported in Walsh et al.[26]. All the details about pre-randomization enrollment into HRZE to test the methods for the six additional individuals that gave informed consent for the pilot pre-randomization phase of the study are also reported in Walsh et al.[26]. Importantly we could not collect stool and blood specimen from one individual from the NTZ arm of the trial and hence only 19 of the 20 NTZ-treated individuals were include in this study's analysis.

*Estimation of* Mtb *load from sputum.* The protocol for estimation of Mtb load from the sputum has been reported in in Walsh et al.[26]. Briefly, *Mtb* was quantified by the number of hours to positive signal using a Bactec 960 automated liquid culture system (Becton and Dickenson [BD], Franklin Lakes, NJ) according to the manufacturer's instructions. This approach has been validated and shown to be comparable to determination of CFU counts[67]. The minimum amount of overnight sputum sample required for *Mtb* quantification was 5 ml, and all overnight sputum samples collected exceeded that amount. Sputum samples were decontaminated via a 20-min incubation, with vortexing, with 5 ml of NALC–NaOH (3% sodium hydroxide, 0.5 to 0.6% N-acetyl-l-cysteine, 1.47% sodium citrate). The endpoint was positive fluorescent signal indicating microbial growth. The reported TTP is the average of the two individual liquid cultures (e.g., technical replicates).

*Six-month longitudinal and observational treatment cohort.* Donors were enrolled through the Clinical Trials Unit at GHESKIO. Pulmonary TB was diagnosed by clinical symptoms, chest radiograph consistent with pulmonary TB, and positive molecular testing. No other formal enrollment criteria were required, like prior antibiotic treatment. All participant samples were deidentified on site using a bar-code system before they were shipped to Weill Cornell Medicine (WCM)/Memorial Sloan Kettering Cancer Center (MSKCC) for analysis. All clinical metadata was collected on site and managed through the REDCap data management system.[68]

*Human healthy controls.* We recruited families of active pulmonary TB patients where at least two siblings within the family were diagnosed with active TB. These criteria were designed to select for households with high risk of transmission of *Mtb*. Household contacts were then recruited if they had been sleeping in the same house with a TB case for at least one month during the six months prior to the TB case diagnosis. Contacts underwent clinical screening for active TB symptoms and IGRA testing. Healthy donors without history of TB contacts or disease were recruited

from the same community as a control group for exposure and also underwent clinical screening for active TB symptoms and IGRA testing. All donors provided informed consent prior to peripheral blood donation for whole blood collection for RNAseq and stool submission for DNA extraction and 16S rDNA sequencing.

**Microbial DNA extraction from stool**. DNA extraction from stool was performed as described.[22] Stool specimens were collected and stored for less than 24 h at 4 °C, aliquoted (~2 ml each), frozen at −80 °C, and shipped to WCM/MSKCC. About 200–500 mg of stool from frozen samples was suspended in 500 µl of extraction buffer (200 mM Tris-HCl [Thermo Fisher Scientific], pH 8.0; 200 mM NaCl [Thermo Fisher Scientific]; 20 mM EDTA [MilliporeSigma]), 210 µl of 20% SDS, 500 µl of phenol/chloroform/isoamyl alcohol (25:24:1; MilliporeSigma), and 500 µl of 0.1-mm–diameter zirconia/silica beads (Biospec Products). Samples were lysed via mechanical disruption with a bead beater (Biospec Products for 2 min, followed by 2 extractions with phenol/chloroform/isoamyl alcohol [25:24:1]). DNA was precipitated with ethanol and sodium acetate at −80 °C for at least 1 h, resuspended in 200 µl of nuclease-free water, and further purified with QIAamp DNA Mini Kit (Qiagen) according to the manufacturer's protocols. DNA was eluted in 200 µl of nuclease-free water and stored at −20 °C.

**16S rDNA sequencing and bioinformatic analysis**. Primers used to amplify rDNA were: 563F (59-nnnnnnnn-NNNNNNNNNNNNN-AYTGGGGYDTAAAGN G-39) and 926R (59-nnnnnnnn-NNNNNNNNNNNNN-CCGTCAATTYHTTTR AGT-39). Each reaction contained 50 ng of purified DNA, 0.2 mM dNTPs, 1.5 µM MgCl2, 1.25 U Platinum TaqDNA polymerase, 2.5 µl of 10× PCR buffer, and 0.2 µM of each primer. A unique 12-base Golay barcode (Ns) preceded the primers for sample identification after pooling amplicons. One to 8 additional nucleotides were added before the barcode to offset the sequencing of the primers. Cycling conditions were the following: 94 °C for 3 min, followed by 27 cycles of 94 °C for 50 s, 51 °C for 30 s, and 72 °C for 1 min, where the final elongation step was performed at 72 °C for 5 min. Replicate PCRs were combined and were subsequently purified using the Qiaquick PCR Purification Kit (Qiagen) and Qiagen MinElute PCR Purification Kit. PCR products were quantified and pooled at equimolar amounts before Illumina barcodes and adapters were ligated on using the Illumina TruSeq Sample Preparation procedure. The completed library was sequenced on an Illumina Miseq platform per the Illumina recommended protocol.

Forward and reverse 16S MiSeq-generated amplicon sequencing reads were dereplicated and sequences were inferred using dada2.[69] Potentially chimeric sequences were removed using consensus-based methods. Taxonomic assignments were made using BLASTN against the NCBI refseq_rna database. These files were imported into R and merged with a metadata file into a single Phyloseq object.

**Peripheral blood transcriptomics**. Collection of peripheral blood and evaluation of host gene expression follows our previous published work[70]. Briefly, peripheral blood was collected into Tempus™ Blood RNA tubes (Applied Biosystems) for the HRZE/NTZ trial cohort, as well as the control cohort. RNA was extracted using the Tempus™ Spin RNA Isolation Kit (Ambion), with addition of on-column DNase treatment (AbsoluteRNA, Ambion). For the EBA longtidunal cohort, PAXgene tubes were used to collect blood according to the manufacturers protocol. RNA was ribo-depleted by polyA selection and libraries were generated using TruSeq (Illumina, San Diego, CA). Paired-end RNA sequences (50 × 50PE) were generated with HiSeq 2500 (Illumina) with at least 50 million reads per sample. Sequence integrity was verified using FastQC (Babraham Bioinformatics). Sequences were aligned to the human genome (version hg38) using STAR aligner[71] and transcript counts were estimated using featurecounts[72]. Quality of aligned and counted reads was assessed using Quality of RNA-Seq ToolSet (QoRTs)[73].

**Statistical and computational analysis**. All statistical and computational analysis was performed in R v.3.6.1 (2019-07-05) with Platform: x86_64-pc-linux-gnu (64-bit), running under: Ubuntu 16.04.6 LTS. The following R packages were used: nlme (v.3.1-141), reshape2 (v.1.4.4), ALEPlot (v.1.1), vita (v.1.0.0), randomForest (v.4.6-14), GSVA (v.1.32.0) msigdbr (v.7.1.1), fgsea (v.1.10.1), Rcpp (v.1.0.5), forcats (v.0.5.0), purrr (v.0.3.4), readr (v.1.3.1), tidyr (v.1.1.2), tibble (v.3.0.4), tidyverse (v.1.3.0), RColorBrewer (v.1.1-2), ggthemes (v.4.2.0), hues (v.0.2.0.9000), edgeR (v.3.26.8), variancePartition (v.1.14.1), scales (v.1.1.1), foreach (v.1.5.1), limma (v.3.40.6), circlize (v.0.4.10), ComplexHeatmap (v.2.0.0), gtools (v.3.8.2), ggplot2 (v.3.3.2), stringr (v.1.4.0), ifultools (v.2.0-5), data.table (v.1.13.0), yingtools2 (v.0.0.0.62), dplyr (v.1.0.2), phyloseq (v.1.28.0), DESeq2 (v.1.24.0), SummarizedExperiment (v.1.14.1), DelayedArray (v.0.10.0), BiocParallel (v.1.18.1), matrixStats (v.0.56.0), Biobase (v.2.44.0), GenomicRanges (v.1.36.1), GenomeInfoDb (v.1.20.0), IRanges (v.2.18.3), S4Vectors (v.0.22.1), BiocGenerics (v.0.30.0).

*Linear mixed effect models*. For the clinical trial, to identify the significance of the influence of sex, age, treatment groups (HRZE and NTZ), and time of treatment on time to positivity (TTP), we implemented a linear mixed effect model as
$TTP \sim Sex + Age + Time * Treatment + 1|ID$. Similarly, to associate the significance of the effect of sex, age, treatment groups (HRZE and NTZ), sequencing batches, and time of treatment on microbiota diversity (Inverse Simpson), Inverse

Simpson was modeled as
$Diversity \sim Sex + Age + Batch + Time * Treatment + 1|ID$, where:

- $1|ID$ is used as a random effect to account for individual differences
- *Sex* represents if an individual is male or female
- *Time* indicates Day 0 and Day 14
- *Treatment* indicates HRZE or NTZ group
- *Batch* represents the sequencing batch information

Alpha diversity indices were computed using phyloseq package in R and the implementation of linear mixed effect models were carried out using nlme package in R.

For the longitudinal observational HRZE treatment cohort, to identify the significance of the influence of sex, age, and treatment time on time to positivity (TTP), we implemented linear mixed effect model as
$TTP \sim Sex + Age + Time + 1|ID$. Similarly, to identify the significance of the influence of sex, age, and treatment time on microbiota diversity on microbiota diversity (Inv Simpson), we performed linear mixed effect modeling
$Diversity \sim Sex + Age + Time + 1|ID$.

Differential analysis for microbial ASVs and host genes: Both raw 16Sr rRNA microbiota ASVs and peripheral blood RNAseq gene-expression counts were modeled using the limma/voom pipeline.[29] This allowed us to use linear mixed-effect modeling of gene/ASV counts as of $Count \sim Sex + Batch + Group + 1|ID$. This model statement enables quantifying sex and sequencing batch-dependent effects in addition to establishing effects that are due to treatment group (pre-treatment, HRZE, NTZ). We included the *Batch* variable as a fixed effect because every treatment group (pre-treatment, HRZE, NTZ) is similarly represented in each sequencing batch. For the six-month longitudinal and observational treatment cohort, we used similar differential analysis approach as the clinical trial by modeling gene/ASV counts as $Count \sim Sex + Batch + Time + 1|ID$, where *Time* represents the Day 0, Day 14, and Day 56 time points. The advantage of limma is that we could use $1|subject$ as a random effect to control for baseline differences among individuals, important in this clinical setting. Significance of ASVs affected by the treatment were determined using a Benjamini–Hochberg false discovery rate (FDR) adjusted *p*-value of 0.05 from the modeling-produced contrast lists (e.g., HRZE vs. Pre, NTZ vs. Pre, Day 0–six-month longitudinal and observational treatment cohort vs. Day 14–six-month longitudinal and observational treatment cohort)[29]. We note that the longitudinal nature of these samples, and having a pretreatment sample allows us to account for other unknown factors (e.g., prior antibiotic use).

To determine how the anti-TB treatment affects both microbiome and peripheral gene expression profiles we performed differential analysis on the counts data obtained by microbiome DNA and peripheral blood RNA sequencing. As the primary endpoint of the clinical trial was powered to determine differences in *Mtb* load (TTP), we determined the statistical power available to identify significant differences in the abundance of both microbiota ASVs, and in the expression of peripheral genes. We ran power calculations to determine that with 16 pre and 16 post treatment microbiome samples and 8 pre treatment and 8 post treatment RNAseq samples for the HRZE cohort, with 80% power at a significance level (α) of 0.05, we could detect a fold change of 1.4 for microbiome difference and a fold change of 1.8 for mRNA transcripts. In the NTZ cohort, with 18 pre treatment and 18 post treatment microbiome samples and 14 pre and 14 post treatment RNAseq samples, with 80% power at α < 0.05, we can detect a fold change of 1.4 for microbiome differences and a fold change of 1.6 for mRNA transcripts. In the longitudinal cohort with 20 baseline, 10 day 7, 18 day 14, 13 one month, 13 two month, and 11 six month follow up microbiome samples, with 80% power at a significance level (α) of 0.05, we can detect a fold change of 1.4 (day 7), 1.36 (day 14), 1.4 (1, 2, and 6 months). In the six-month longitudinal and observational treatment cohort with 19 baseline, 19 day 14, and 13 two-month RNAseq samples, with 80% power at a significance level (α) of 0.05, we can detect a fold change of 1.4 for mRNA transcripts at day 14, and 1.5 at two months. Power calculations were performed with the RNAseqPower package in R. For microbiome data we calculated a biological coefficient of variation of 0.3, and for RNAseq, we used a coefficient of variation of 0.4. We estimated the expected minimum fold change that we could observe for each group based on the sample size, sequencing depth, and an α of 0.05. To visualize trends of transcript fold changes, we used scatter plots and calculated post-vs-pre fold changes for all transcripts throughout.

*Host-Microbiome-Mtb modeling*. We assessed the relative contribution of the gastrointestinal microbiota and *Mtb* dynamics towards peripheral gene expressions using Random forest regression (RFR). Instead of modeling each gene/transcript profile as a function of microbiome and TTP, we mapped our gene expression data to a set of 50 Hallmark Pathways via GSVA. To avoid having correlated samples from same individual in a model, we instead modeled the changes in normalized enrichment score of Hallmark pathways at two time points (Day 0 and Day 14) as a function of change in microbiome and TTP at corresponding timepoints. For each Hallmark pathway *l* identified to be significantly affected by HRZE (HRZE/NTZ trial or six-month longitudinal and observational treatment cohort) or NTZ via linear mixed effects modeling, we first computed the change in normalized enrichment score $\Delta NES^l$ between two consecutive time points *Day 0* and *Day 14* as $\Delta NES^l = NES^l_{14} - NES^l_0$ for each individual. We then regressed this quantity against the corresponding Log2 fold change of normalized counts value (NEV) of every ASV *v* as $Log2(NEV^v_{14}/NEV^v_0)$ in

the same interval and against the log2 fold change in TTP i.e., $Log2 (TTP_{14}/TTP_0)$. Normalized expression value (NEV) is the CPM (counts per million) obtained by normalizing the raw counts by the library sizes and multiplying by one million. To train the models we used observations from the clinical trial and from the longitudinal EBA cohort for a total of 34 paired samples. We fit a model for each significant pathway using all the data from the three patients' group (HRZE clinical trial, NTZ clinical trial, and HRZE EBA) because we wanted to find patterns that are general across multiple datasets. Each model was trained using 5000 generated trees and a train-validation partitioning of 80–20% of the data.

Permutated importance[46] measure is used to assess the importance and significance of predictors (e.g., TTP, microbiome constituents, etc.) towards the dependent variable (e.g., pathways). The higher the permutation importance of a predictor, the higher is the association towards the outcome variable. Accumulated Local Effects (ALE) plots[47] are used to estimate the relationship between the predictors towards the outcome variable. To simplify the effect of predictors on pathways into a monotonic relationship, we computed the slope of a fitted straight line of ALE plots and summarize the direction of the slope into a positive/negative effect of predictors towards outcome variable.

Random Forest Regression Analysis of control cohort: To investigate microbiome-pathway relationships in the FC and CC cohort, we modeled the normalized enrichment score (NES) of each Hallmark pathway as a function of normalized expression value (NEV) of ASVs for corresponding samples. Each RFR model was trained using 5000 generated trees and a train-validation partitioning of 80–20% of the data. Permutated importance[46] and (ALE) plots[47] were used to assess the importance and relationship of ASVs towards pathways.

Within sample GSEA analysis: The ssGSEA (single sample gene set enrichment analysis) method[74] was used to profile within-sample differences between pathways from the MiSigDB Hallmark pathways list[38], or other MiSigDB lists (e.g., KEGG), with the GSVA package in R[55]. The MiSigDB Hallmark pathway list is a well validated set of general curated biological pathways that give insight into specific biological and cellular processes. In addition, we obtained a list of well validated active TB signatures from the TBSignatureProfiler R package (https://github.com/compbiomed/TBSignatureProfiler). Variance stabilized transformed (vst) counts derived from DESeq2 were used as input into the GSVA function in the GSVA R package with default parameters (kcdf = "Gaussian") and scaled Normalized Enrichment Scores (NES) were plotted as heatmaps. Importantly, unlike classical GSEA, this analysis is agnostic to sample phenotype.

Identification of differential pathways post antibiotic treatment in both the HRZE/NTZ trial and longitudinal observational HRZE treatment cohort was performed using linear mixed effect model where we modeled the normalized enrichment score (NES) of each pathways as $NES \sim sex + batch + group + 1|ID$ and $NES \sim sex + batch + Time + 1|ID$, respectively. Significance of pathways affected by the treatment were determined using a Benjamini–Hochberg false discovery rate (FDR) adjusted p-value of <0.05. Significant pathways in both trials were used as outcome/dependent variables for the RFR models.

To finally confirm that HRZE treatment was renormalizing peripheral inflammatory pathways while NTZ was causing exacerbation we applied mixed effect modeling to predict the pairwise distance in NES between every CC/FC sample and every sample before or after treatment from the HRZE/NTZ trial and the longitudinal observational HRZE treatment cohort independently. In the first case we fit the model $Distance \sim sex + batch + group + 1|ID$ in the second we fit $Distance \sim sex + batch + Time + 1|ID$. P-value and sign of the coefficient associated with $group$.

**Reporting summary**. Further information on research design is available in the Nature Research Reporting Summary linked to this article.

## Data availability

Data on Time to Positivity where obtained from Walsh et al.[26] and are available on Github at https://wipperman.github.io/TBRU/. 16S rDNA sequencing data is deposited with the SRA under accession no. PRJNA445968 (https://www.ncbi.nlm.nih.gov/bioproject/PRJNA445968). Peripheral blood transcriptomic data are deposited with the SRA under accession no. PRJNA445968 (https://www.ncbi.nlm.nih.gov/bioproject/PRJNA445968). All the results from the machine learning and statistical calculations are available as Supplementary Data. All the metadata and code to analyze the data presented in this manuscript and to reproduce all of the figures and results is available on Github at https://wipperman.github.io/TBRU/TB_paper_2020/. Source data are provided with this paper.

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

## Acknowledgements

M.F.W., C.K.V., Y.T., K.W., C.N., D.W.F., M.S.G. acknowledge funding from the Tri-I TBRU (grant: U19AI111143). C.V. acknowledges support from K08AI132739. M.F.W. acknowledges support from the National Center for Advancing Translational Sciences (grant: TL1TR002386-02). Support for NCT02684240 came primarily from the Abby and Howard P. Milstein Program in Chemical Biology and Translational Medicine. V.B. acknowledges support from the National Science Foundation (grant: 1458347). This work was supported by P30 CA008748.

## Author contributions

Patient recruitment, enrollment, and sample collection were contributed by L.M., K.M., K.F.W., J.B., and S.C.V.; Laboratory experiments were performed by M.F.W. and C.K.V.; Data analysis was performed by M.F.W., S.K.B., V.B.; Machine learning modeling was performed by S.K.B., V.S.M., and V.B. Wrote manuscript: M.F.W., S.K.B., M.S.G., V.B.; Edited manuscript: M.F.W., S.K.B, C.K.V., V.S.M., Y.T., L.M., K.M., S.C.V., D.F., J.B., K.F.W., C.N., D.W.F., M.S.G., V.B. M.F.W. and S.B. are co-first authors, and M.S.G. and V.B. are co-last authors. Co-authorship and author order were determined by recognition that the integration of the nuances clinical trial data and mathematical modeling are different skillsets found in different laboratory environments. Each were important components to the validity and message of this manuscript.

## Competing interests

M.F.W. is currently an employee and shareholder of Regeneron Pharmaceuticals, Inc. M.S.G. reports consulting fees and equity in Vedanta Biosciences, Inc., consulting fees from Takeda, and is on the SAB of PRL-NYC. V.B. is supported by a Sponsored Research Agreement from Vedanta Biosciences, Inc. The remaining authors declare no competing interests.
