## [Peer Review File · Nature Communications]

REVIEWER COMMENTS

Reviewer #3 (Remarks to the Author):

I would like first to acknowledge the substantial effort the Authors have invested in implementing my suggestions. I am well aware how much work that was and I am very satisfied by the outcome.

I think of the major issues only one thing remains: is the observed inflammatory exacerbation in the NTZ arm due to "microbiome-dependent immunity that could be responsive to the observed antimicrobial perturbations" or (at least in part) due to the prolonged effect of the disease? As I have mentioned in the previous review, regressing using the relative differences (as described on p. 17) resolves only a part of the problem of the causality here (which I described at length previously). In short, to reformulate my objections, the Authors make an implicit assumption that TTP (a measure of bacterial load, known to correlate with disease severity, the only measure of disease state or severity used by the Authors) is sufficient as a predictor variable to account for the effect that the disease has on the systemic immune response. That is, that the TTP perfectly reflects the changes in the immune responses which are caused by the 2 week exposure to untreated TB (irrespective of the NZT).

This is almost certainly not true – despite lack of observed increase in bacterial load (which can be limited by other factors), the state of the organism can change. Correlation (between TTP and disease severity) does not indicate identity or lack of other hidden variables. For example, we know from mouse models that while CFU is correlated to directly observed pathology, it is by no means a perfect predictor. The same is true for parameters such as clinical disease score or pulmonary destruction observed in humans.

While I concur with the Authors that much of what they present seems to support their view (ie., the role of the microbiome), and that presently I am convinced that at least the TTP is a poorer predictor than the microbiome, and that overall their argumentation is solid, at minimum they should also mention and discuss this possibility. If the Authors think that I am on a completely wrong track here, maybe they could explain shortly why in the Discussion, as I think other TB researchers might also wonder at an apparent lack of effect of 2 weeks of untreated disease on transcriptomic responses (as opposed to the effect of microbial depletion by NZT).

Minor issues:

* I think it would be better to refer to NTZ and HRZE groups consistently as "arms" (as it is used predominantly) or groups rather than cohorts (as it is in the legend to fig. 4, for example) or on p. 9, l. 212. Using "cohort" in this context is confusing, since Fig. 1 introduces the three cohorts for the study, and the "cohorts" referred in Fig 4 and on p. 9 correspond to the two arms of cohort 1 from Fig 1.

* "sequencing batch information has been controlled for by including 199 it as fixed effect in the modeling" – I am aware of the technical difficulties that entails, and I am adding this only pro forma, but as a rule batch effects are considered to be random effects.

* P. 18, l. 333/334: "Interestingly, while NTZ affected a smaller subset of these IBD 334 immune genes, it caused greater exacerbation compared to HRZE (77.8% vs 42.0% genes exacerbating)"

I think that one could easily claim the opposite, stating that the effect sizes (log 2 fold changes) in case of IBD exacerbating genes were much lower than either in case of IBD/NZ renormalizing genes or IBD/NZ exacerbating genes. I don't think this statement is truly backed up by data.

Kind regards,

January Weiner

Reviewer #4 (Remarks to the Author):

The manuscript by Wipperman et al. sets out to address an important obstacle in human gut microbiome studies, namely directly connecting a disease phenotype/outcome with changes to the microbiome. To pursue this overarching goal, they take advantage of 2 patient cohorts in Haiti receiving anti-TB drug treatment and compare treatment outcomes to changes in microbiome and peripheral immune gene signature. The longitudinal analysis is a particular strength. Overall, the manuscript is very well-written, thorough and innovative. Furthermore, after reading the reviews from their prior submission to [redacted] this manuscript has been substantially improved. I particularly appreciated the cross-referencing of their immune gene dataset to previously published TB and IBD studies. Although this study does not demonstrate causality between microbiome changes, immune signatures and TB disease outcomes, it provides creative analysis of unique cohorts and their relationship with other independently-generated cohorts. Further, their results substantiate published, mechanistic animal studies. These advancements without defined functional studies make this work appropriate for Nat Comm. However, I have a couple comments that should be addressed to clarify the significance of their results:

Major comments:

1. In line 246, the authors state that "Additionally, NTZ caused an expansion of known oxygen-tolerant pathobionts 246 including, *K. pneumoniae*, *E. coli*, *C. freundii*, *S. alactolyticus*, and *E. faecium*". This difference appears to be the most striking difference between HRZE and NTZ-treated groups. This was investigated more closely in Figure 6. However, in its current form, I found the depiction of this relationship confusing. Is there any single or group of oxygen-tolerant pathobionts that is predictive of peripheral inflammation and TTP outcomes or is their only an association when you also consider the changes in Clostridia? A more clear explanation and/or depiction of this result would be informative.
2. Comparison to other TB trial and IBD patient datasets was clever and informative. I also appreciate provision of the complete datasets. However, I would like to see spreadsheets or lists that highlight (or are limited to) the specific genes the authors are referring to in the text related to figure 4.

Reviewer #5 (Remarks to the Author):

Major comments:

The microbiome and TB is a major neglected research area. The finding that specific stool taxa may, in concert with treatment, exert a defined effect on the host transcriptional phenotype is important and novel. To show this, Wipperman et al. used Random Forest Regression models to correlate stool microbiota changes, sputum Mtb TTP (a measure of drug regimen kill or treatment response) and host peripheral gene expression across three datasets. The authors found that active tuberculosis patients randomised to HRZE or NRT had, despite differences in Mtb kill, a perturbed gut microbiota (specifically Clostridiales spp.) that correlated with inflammatory pathways. Overall, the manuscript is written in a clear manner and inclusion of NTZ to demonstrate treatment-specific changes a strength. The use of per individual paired analyses are a strength.

- 1) As requested, I first reviewed responses to the prior reviewer comments.
 - a. This process would have been easier if the authors pasted the relevant amended text below each response. For some important comments (e.g., reviewer #1's first comment) it is unclear

whether the authors' response, which is well written and justified, was added to the manuscript. I suggest elements of each response are always added to the manuscript, and this is made clear – otherwise readers who do not see the responses will just have the same queries as the reviewers. I ask the authors to confirm they have done this.

b. Although rightly raised by the prior reviewers, the language with regard to "prediction" and "role" and "causation" rather than "association" and "correlation" still requires softening.

"Prediction" also suggests a completely independent validation exercise, which is not entirely true (they authors appear to have used an 80/20 split, and a healthy cohort for validation). There has arguably therefore not been external validation (and healthy controls have well known limitations). Furthermore, such cross-validation approaches can be prone to overfitting. The Discussion is especially loose with this language.

c. The causal pathway requires careful unpacking. Treatment kills Mtb which in turn reduces proinflammatory host immune signatures. This is well known already. Treatment also kills other bacteria that allow the Clostridia and other taxa to increase. This is also known already. The authors now show the microbiome may be linked to the transcriptome. This does not mean that these taxa themselves reduce pro-inflammatory signatures nor that other mechanisms are not at play. This should be more carefully described in the absence of carefully controlled laboratory and animal experiments. Other pathways should be described. I see this has been raised by more than one other reviewer.

d. That said, I do not think the absence of such experiments or functional assays is critical. The findings provide justification for such experiments in future work, and that alone is already a significant contribution.

e. Overall, the manuscript appears to be significantly strengthened compared to the prior submission.

f. The methodology is also strengthened as the authors have discovered and adjusted batch-to-batch variation in their analyses.

2) A major limitation of the work is that people with signs and symptoms of TB, but who did not have TB, are not included. In other words, one cannot tell if any baseline differences in the microbiome (and the associations seen with the transcriptome at this point) are necessarily unique to TB.

3) The authors make several conclusions about the "renormalisation" of the inflammatory host immune response but, as they do not have a specimen from patients before perturbation of their immune system due to TB disease, how can they conclude that it has "re-normalised"? Rather, they should just simply say it more closely resembles a transcriptome seen in separate healthy controls from a different cohort. As the authors acknowledge, this different cohort is very different to TB patients.

4) There is also no inclusion of any type of respiratory specimen. TB is primarily a respiratory disease.

5) The first finding in the abstract – that TB treatment reduces immune signatures characteristic of active TB, is extensively documented in the literature

(<https://www.ncbi.nlm.nih.gov/pmc/articles/PMC5658513/> for example). The current authors also found a less effective regimen to reduce signatures less. This finding is not novel and I suggest the authors focus their abstract elsewhere (the microbiome and its relationship with the transcriptome).

6) There are other significant aspects that must be improved before publication. For example, in many instances (i.e. alpha- and beta-diversity changes during treatment), actual p-values and the strength of correlation between taxa and genes are lacking. These statistics are required for the results to support the conclusions, and convince the reader that this is indeed the case.

Other comments

1. Abstract (lines 40-66): The language could be simplified so that the main messages are clearly described.

2. Abstract (short): "tissues" appears to be the incorrect word.

3. The NTZ regimen is not used routinely, which limits the applicability of findings. This requires acknowledge and perhaps some explanation for the reader as to why NTZ may still be valuable as a research tool.

4. Introduction

a) Lines 101 and 103: "infection by Mycobacterium tuberculosis" should be changed to active TB disease, unless the authors intend to refer to latent infection.

- b) Line 109: important to mention rifampicin is broad spectrum.
 - c) It is odd to refer to figures in the Introduction. I am not sure if the journal allows this.
 - d) The intro is very long (the final paragraph is a full page).
5. The meta-analysis of RNA signatures and TB outcomes is not the best reference. Far richer and rigorous, and better validated and published signatures from MAs have been published, such as <https://www.ncbi.nlm.nih.gov/pmc/articles/PMC4838193/> and should rather be used. Furthermore, the Berry study is often incorrectly referred to as a meta-analysis in the manuscript text.

6. Results

- a) Line 202: Why is Shannon diversity mentioned here?
- b) Line 348-353: Please cite exact p-values for comparisons between time points in Figure 5A and B.
- c) Line 466: 18 FC and 36 CC do not add up to 52.
- d) Rephrase using more formal language: "While these pathways ..., we do feel that ... disease" in lines 473-5 and "microbiome space" in line 478.
- e) "Intestinal microbiome", "gastrointestinal microbiome", "gut microbiota" are all used interchangeably. Authors should stick to one term throughout. More accurately, they measured the stool microbiome – not the (gastrointestinal) microbiome. The title should reflect this.

7. Figures

Figure 1

- a. Rather add an extra column for TTP in B and remove A and C as these sections do not contain information not already presented in B.
- b. In the legend for (A), indicate what HRZE and NTZ stand for. "TB-negative" should be changed to something more specific (i.e. IGRA-negative). In (C), the word "causal" is overly ambitious.

Figure 2

- a. Fix spelling for "pretreatment" in (A).
- b. Colours for HRZE and NTZ are difficult to distinguish.
- c. In (B), quote exact p-values and state TTP is measured in hours on y-axis.
- d. Annotate three symbols at bottom right of (C). and include PERMANOVA results.
- e. X-axes in (B) and (D) should be only include days 0 and 14.

Figure 3

- a. The colours in the volcano plot are difficult to distinguish.
- b. It would be beneficial to the reader to annotate the most differentially abundant taxa - at minimum those from the order Clostridiales in Figure 3A.
- c. Indicate clearly on the plot which timepoint is represented by a log-fold change <0 and which is >0. This applies to all volcano plots in the manuscript.

Figure 4

- a. Indicate what "NES" stands for.

Figure 6

- a. "Blue/orange entries indicate features found to significantly associate with changes in a specific inflammatory pathway," however, the strength of this association (or statistical significance i.e. p-value or correlation coefficient with CI) is not described.
- b. Similarly, can this be expressed statistically for "Black dots are used to identify the top important predictor"?

Figure 7

- a. Delete the colourful legend to the right in (A) - it is redundant with the top labels.
- b. Groups should be arranged according to respective cohorts (i.e. EBA groups alongside each other)

8. Methods

- a) Were study participants on any antibiotics prior to study enrolment? A trial by antibiotics is common in many settings prior to patients commencing TB treatment.
- b) Were family contacts and community controls included in the study if both asymptomatic and IGRA-negative?

RESPONSE TO REVIEWERS COMMENTS

REVIEWER COMMENTS

Reviewer #3 (Remarks to the Author):

I would like first to acknowledge the substantial effort the Authors have invested in implementing my suggestions. I am well aware how much work that was and I am very satisfied by the outcome.

Response: We want to again thank the reviewer for reviewing our manuscript the first time, for providing really insightful comments and suggestions, and for agreeing to review the manuscript a second time. We are really happy to know that the reviewer appreciated the work that we did to address the comments and that he/she is satisfied with the outcome of our revision. We have addressed all the additional comments from this review as well (see points below).

I think of the major issues only one thing remains: is the observed inflammatory exacerbation in the NTZ arm due to "microbiome-dependent immunity that could be responsive to the observed antimicrobial perturbations" or (at least in part) due to the prolonged effect of the disease? As I have mentioned in the previous review, regressing using the relative differences (as described on p. 17) resolves only a part of the problem of the causality here (which I described at length previously). In short, to reformulate my objections, the Authors make an implicit assumption that TTP (a measure of bacterial load, known to correlate with disease severity, the only measure of disease state or severity used by the Authors) is sufficient as a predictor variable to account for the effect that the disease has on the systemic immune response. That is, that the TTP perfectly reflects the changes in the immune responses which are caused by the 2 week exposure to untreated TB (irrespective of the NZT).

This is almost certainly not true – despite lack of observed increase in bacterial load (which can be limited by other factors), the state of the organism can change. Correlation (between TTP and disease severity) does not indicate identity or lack of other hidden variables. For example, we know from mouse models that while CFU is correlated to directly observed pathology, it is by no means a perfect predictor. The same is true for parameters such as clinical disease score or pulmonary destruction observed in humans.

While I concur with the Authors that much of what they present seems to support their view (ie., the role of the microbiome), and that presently I am convinced that at least the TTP is a poorer predictor than the microbiome, and that overall their argumentation is solid, at minimum they should also mention and discuss this possibility. If the Authors think that I am on a completely wrong track here, maybe they could explain shortly why in the Discussion, as I think other TB researchers might also wonder at an apparent lack of effect of 2 weeks of untreated disease on transcriptomic responses (as opposed to the effect of microbial depletion by NZT).

Response: We agree with the reviewer that we cannot exclude a non-microbiome effect explaining the exacerbations in the NTZ group. We acknowledge this caveat in the discussion. We also recognize that we did not measure other host parameters to quantify state of the volunteers apart from TTP, and agree that there are likely other host parameters (e.g., white blood cell count, T cell functionality assays) in addition to TTP/CFU and microbiome that could altogether increase ability to predict the exacerbation state of NTZ-treated individuals. As suggested by the reviewer we also discuss this as caveat of the study in the Discussion section

(see page 28). We also point out that the paper's objective is to demonstrate associations between microbiome dynamics, pathogen load and host inflammatory state using three data modalities (rDNA sequencing, sputum TTP and blood RNAseq) that can be readily and rapidly measured from humans (or in the field). We also point this out in the paper (see page 28).

Minor issues:

* I think it would be better to refer to NTZ and HRZE groups consistently as "arms" (as it is used predominantly) or groups rather than cohorts (as it is in the legend to fig. 4, for example) or on p. 9, l. 212. Using "cohort" in this context is confusing, since Fig. 1 introduces the three cohorts for the study, and the "cohorts" referred in Fig 4 and on p. 9 correspond to the two arms of cohort 1 from Fig 1.

Response: We agree with the reviewer that our descriptions of the cohorts was confusing. We have comprehensively revised this terminology such that the HRZE and NTZ are arms of the same clinical trial (HRZE/NTZ trial), and made sure the paper is throughout consistent.

* "sequencing batch information has been controlled for by including 199 it as fixed effect in the modeling" – I am aware of the technical difficulties that entails, and I am adding this only pro forma, but as a rule batch effects are considered to be random effects.

Response: We agree with the reviewer that batch is often modeled using a random effect and not fixed. However, it is also acknowledged that multiple sources of random effects make mixed-effect models really hard to fit, and as such several modeling frameworks to perform regression analysis with counts data (e.g. edgeR, DESeq2) either treat batches only as fixed effects (because they do not allow for random effects at all) or only allow to model a single random effect (e.g. limma/voom).

See also: <https://www.sciencedirect.com/science/article/pii/S200103701930409X#b0050>. Specifically, to our study, every treatment group is represented in every batch (see table below), therefore we find appropriate to include batch as an additional fixed effect to control for it.

	pool749	pool863	pool864
Post HRZE	7	2	7
Post NTZ	3	4	9
Pre HRZE	7	2	7
Pre NTZ	3	5	10

We added the rationale underlying our choice in the Methods section, under Statistical and Computational Analysis.

* P. 18, l. 333/334: "Interestingly, while NTZ affected a smaller subset of these IBD 334 immune genes, it caused greater exacerbation compared to HRZE (77.8% vs 42.0% genes exacerbating)"

I think that one could easily claim the opposite, stating that the effect sizes (log 2 fold changes) in case of IBD exacerbating genes were much lower than either in case of IBD/NZ renormalizing genes or IBD/NZ exacerbating genes. I don't think this statement is truly backed up by data.

Response: After carefully reviewing the statement and the data we agree with the Reviewer and removed the statement.

Reviewer #4 (Remarks to the Author):

The manuscript by Wipperman et al. sets out to address an important obstacle in human gut microbiome studies, namely directly connecting a disease phenotype/outcome with changes to the microbiome. To pursue this overarching goal, they take advantage of 2 patient cohorts in Haiti receiving anti-TB drug treatment and compare treatment outcomes to changes in microbiome and peripheral immune gene signature. The longitudinal analysis is a particular strength. Overall, the manuscript is very well-written, thorough and innovative. Furthermore, after reading the reviews from their prior submission to [redacted], this manuscript has been substantially improved. I particularly appreciated the cross-referencing of their immune gene dataset to previously published TB and IBD studies. Although this study does not demonstrate causality between microbiome changes, immune signatures and TB disease outcomes, it provides creative analysis of unique cohorts and their relationship with other independently-generated cohorts. Further, their results substantiate published, mechanistic animal studies. These advancements without defined functional studies make this work appropriate for Nat Comm. However, I have a couple comments that should be addressed to clarify the significance of their results:

Response: We want to thank the reviewer for appreciating our work. We are very happy to know that the reviewer finds our work to address an important question in human microbiome studies. It is also rewarding to know that the reviewer appreciated the use of the patient cohorts and the longitudinal nature of our data to identify associations between microbiome, host transcriptomics, and TB disease. We are happy to see that the reviewer acknowledged our ability to address the comments from our previous revision at [redacted] and that he/she thinks that even though we cannot demonstrate causality, the identified associations confirm a body of work from mechanistic (animal studies) in humans and our analysis was creative to connect findings from multiple cohorts and types of data. We are happy to see that the reviewer deems our study appropriate for publication in *Nature Communications*. Below is our response to all reviewers' comments.

Major comments:

1. In line 246, the authors state that "Additionally, NTZ caused an expansion of known oxygen-tolerant pathobionts 246 including, *K. pneumoniae*, *E. coli*, *C. freundii*, *S. alactolyticus*, and *E. faecium*". This difference appears to be the most striking difference between HRZE and NTZ-treated groups. This was investigated more closely in Figure 6. However, in its current form, I found the depiction of this relationship confusing. Is there any single or group of oxygen-tolerant pathobionts that is predictive of peripheral inflammation and TTP outcomes or is their only an association when you also consider the changes in Clostridia? A clearer explanation and/or depiction of this result would be informative.

Response: We thank the reviewer for this comment. As the reviewer pointed out *K. pneumoniae*, *E. coli*, *C. freundii*, *S. alactolyticus*, and *E. faecium* are all enriched after NTZ treatment. When we model changes in host pathways as a function of changes in the microbiome via Random Forest Regression, all species that are significantly affected by the antibiotic treatments are included as predictors. The model selects the species that significantly contribute to a certain host pathway by using permuted importance analysis (See Methods). As depicted in Figure 6, all these oxygen-tolerant pathobionts (with the exception of *K. pneumoniae* and *C. freundii*) are found to have a positive impact on a number of inflammation-related pathways (or gene sets) while "controlling" for the abundance of all other modeled bacteria. If the association between abundance of these bacteria and inflammation was just a result of the loss of Clostridia these bacteria would not be have been identified by the model as

important predictors because explicitly modeling Clostridia would push them out of significance. Instead, the model finds the abundance of these bacteria to aid significantly higher predictive ability than just modeling pathways as a function of Clostridia species alone. We elaborate this more in the paper (see page 19).

2. Comparison to other TB trial and IBD patient datasets was clever and informative. I also appreciate provision of the complete datasets. However, I would like to see spreadsheets or lists that highlight (or are limited to) the specific genes the authors are referring to in the text related to figure 4.

Response: As the reviewer suggested we created an Excel file in the Supplementary Data containing the list of all genes included in the TB signature, all the list of genes included in the IBD signature. For each gene we also tabulate and report corresponding p value and the foldchange displayed in Figure 4B-E. In the same Excel file, we also report the list of genes corresponding to each pathway reported in Figure 4A. We added a statement to Figure 4 caption to indicate availability of this data as Supplementary Data.

Reviewer #5 (Remarks to the Author):

Major comments:

The microbiome and TB is a major neglected research area. The finding that specific stool taxa may, in concert with treatment, exert a defined effect on the host transcriptional phenotype is important and novel. To show this, Wipperman et al. used Random Forest Regression models to correlate stool microbiota changes, sputum Mtb TTP (a measure of drug regimen kill or treatment response) and host peripheral gene expression across three datasets. The authors found that active tuberculosis patients randomised to HRZE or NRT had, despite differences in Mtb kill, a perturbed gut microbiota (specifically Clostridiales spp.) that correlated with inflammatory pathways. Overall, the manuscript is written in a clear manner and inclusion of NTZ to demonstrate treatment-specific changes a strength. The use of per individual paired analyses are a strength.

Response: We want to thank the reviewer for evaluating our paper and the response to the comments to our previous submission to [redacted]. We are happy to see that the reviewer deems that our findings on how specific microbiome components in concert with treatment affect host transcriptional response important and novel. We are also happy to see that the reviewer finds our manuscript clearly written and determine the per-individual paired analysis another strength of the work. Below is a point-by-point response to the reviewer comments.

1) As requested, I first reviewed responses to the prior reviewer comments.

a. This process would have been easier if the authors pasted the relevant amended text below each response. For some important comments (e.g., reviewer #1's first comment) it is unclear whether the authors' response, which is well written and justified, was added to the manuscript. I suggest elements of each response are always added to the manuscript, and this is made clear – otherwise readers who do not see the responses will just have the same queries as the reviewers. I ask the authors to confirm they have done this.

Response: We apologize that this was not clear from our first revision. We confirmed that we addressed all responses to the first set of reviewers' comments from the [redacted] round of reviews with specific additions and edits in different part of the paper.

b. Although rightly raised by the prior reviewers, the language with regard to “prediction” and “role” and “causation” rather than “association” and “correlation” still requires softening. “Prediction” also suggests a completely independent validation exercise, which is not entirely true (the authors appear to have used an 80/20 split, and a healthy cohort for validation). There has arguably therefore not been external validation (and healthy controls have well known limitations). Furthermore, such cross-validation approaches can be prone to overfitting. The Discussion is especially loose with this language.

Response: We agree with the reviewer and softened the language accordingly throughout the whole paper. We additionally state in the Discussion section (see page 28) that future study design can address questions of causality: “Whether these relationships are causal, or biomarkers of another state will remain at the forefront of future study design.” However, we do not understand the reviewer comment of “and healthy controls have well known limitations”, because it is a completely independent cohort which is not used in model training (e.g. no data leakage) and once it is modeled independently it displays overlapping associations with those obtained when modeling the longitudinal data. Hence, we firmly believe it to be a true, independent validation.

c. The causal pathway requires careful unpacking. Treatment kills Mtb which in turn reduces proinflammatory host immune signatures. This is well known already. Treatment also kills other bacteria that allow the Clostridia and other taxa to increase. This is also known already. The authors now show the microbiome may be linked to the transcriptome. This does not mean that these taxa themselves reduce pro-inflammatory signatures nor that other mechanisms are not at play. This should be more carefully described in the absence of carefully controlled laboratory and animal experiments. Other pathways should be described. I see this has been raised by more than one other reviewer.

Response: We agree that this study was not designed to detect latent variables or some unknown confounding factor, a variable that is related to both change in microbiome state as well as change in peripheral gene signatures, for example, but is not itself either of these things. We have elaborated more in the discussion section to explicitly refer this as one of the caveats of our study (see page 28).

d. That said, I do not think the absence of such experiments or functional assays is critical. The findings provide justification for such experiments in future work, and that alone is already a significant contribution.

Response: We are happy to see that the reviewer finds our work important in providing justification for future work and considers it a significant contribution. We added some text to place this paper into broader context too (see page 28 and Response to Comments #1 from Reviewer #1).

e. Overall, the manuscript appears to be significantly strengthened compared to the prior submission.

Response: We are happy to see that reviewer noted this.

f. The methodology is also strengthened as the authors have discovered and adjusted batch-to-batch variation in their analyses.

Response: We are happy to see that reviewer noted this.

2) A major limitation of the work is that people with signs and symptoms of TB, but who did not have TB, are not included. In other words, one cannot tell if any baseline differences in the microbiome (and the associations seen with the transcriptome at this point) are necessarily unique to TB.

Response: We have to respectfully disagree with the Reviewer here and do not consider this as limitation of the work. The actual strength of our work is the fact that we are comparing each person to their own pretreatment sample, such that the changes in microbiome and transcriptome that occur during intervention are controlled for each person.

3) The authors make several conclusions about the “renormalisation” of the inflammatory host immune response but, as they do not have a specimen from patients before perturbation of their immune system due to TB disease, how can they conclude that it has “re-normalised”? Rather, they should just simply say it more closely resembles a transcriptome seen in separate healthy controls from a different cohort. As the authors acknowledge, this different cohort is very different to TB patients.

Response: We agree that by not having a pre-TB sample for this people we cannot say that these people are returning to their pre-TB state. However, we define renormalization as seeing a treatment/pre-treatment fold change having the same sign as the Control/Active TB fold change for the 373 extensively validated genes from the Berry study (reference 19). Moreover, we have added new linear-mixed modeling analysis showing that HRZE treatment (but not NTZ) significantly push the peripheral inflammatory response closer to that of the community control and family contacts cohort (see page 23). We believe that this control cohort (Cohort 3, N=55) of family contacts and community controls is an accurate representative sample of individuals in Haiti (pre-TB). These individuals do indeed represent a normal inflammatory state (roughly half are IGRA negative and half IGRA positive).

4) There is also no inclusion of any type of respiratory specimen. TB is primarily a respiratory disease.

Response: This is true, and we do not study or collect respiratory specimens in this study. In fact, this is an invasive and costly procedure, prone to contamination and noise (requiring larger sample sizes), and difficult to perform in resource limited settings like Haiti. However, we are familiar with the respiratory microbiome literature, and do not think that any data presented in this paper are challenged (and are in fact supported) by the claims we make here.

5) The first finding in the abstract – that TB treatment reduces immune signatures characteristic of active TB, is extensively documented in the literature (<https://www.ncbi.nlm.nih.gov/pmc/articles/PMC5658513/> for example). The current authors also found a less effective regimen to reduce signatures less. This finding is not novel, and I suggest the authors focus their abstract elsewhere (the microbiome and its relationship with the transcriptome).

Response: We reworded the abstract to prioritize and focus it on the transcriptome-microbiome relationships.

6) There are other significant aspects that must be improved before publication. For example, in many instances (i.e. alpha- and beta-diversity changes during treatment), actual p-values and the strength of correlation between taxa and genes are lacking. These statistics are required for the results to support the conclusions and convince the reader that this is indeed the case.

Response: We appreciate the reviewer comments. We did create a supplementary file with all that information. As first submission it was not included in the pdf file that was transferred by the journal. We now provide an Excel file that has (1) p-values, adjusted p values, and effect sizes for all Limma/Voom microbiome and host genes differential analysis, (2) p-values, adjusted p values, and effect sizes for linear-mixed effect modeling for TTP and diversity, (3) p-values, adjusted p values, and effect sizes for linear-mixed effect modeling for the host pathways, (4) p-values associated with the permuted importance modeling from the microbiome-pathways analysis.

Other comments

1. Abstract (lines 40-66): The language could be simplified so that the main messages are clearly described.

Response: We have reworded the text in the abstract.

2. Abstract (short): "tissues" appears to be the incorrect word.

Response: We agree, and this statement was altered to: "To do this we applied machine learning methods for high-dimensional data to learn associations in humans between these commonly profiled human samples".

3. The NTZ regimen is not used routinely, which limits the applicability of findings. This requires acknowledge and perhaps some explanation for the reader as to why NTZ may still be valuable as a research tool.

Response: We agree that the applicability of NTZ as a disruptor of the microbiome in the context of TB treatment is (likely) not applicable, however in our study NTZ happens to act as a control as it does not kill Mtb but perturbs the microbiome. By including the NTZ data in the modeling we are now able to decouple associations that Mtb and microbiome have on peripheral immunity. We reinforce this concept on page 17.

4. Introduction

a) Lines 101 and 103: "infection by *Mycobacterium tuberculosis*" should be changed to active TB disease, unless the authors intend to refer to latent infection.

Response: This was altered to: "Infection by *Mycobacterium tuberculosis* (Mtb) with active TB disease..."

b) Line 109: important to mention rifampicin is broad spectrum.

Response: We have updated the text to reflect this statement, however, RIF is not truly "broad spectrum": "Antibiotic treatment for active TB involves combination therapy with narrow (HZE), and semi-broad (R) spectrum, and prodrug (HZ) agents with mostly *Mycobacterial*-specific targets, (HRZE)..."

c) It is odd to refer to figures in the Introduction. I am not sure if the journal allows this.

Response: Placing figures in the introduction has been done previously in Nature Communications (e.g., <https://www.nature.com/articles/s41467-020-18834-6> & <https://www.nature.com/articles/s41467-020-18853-3>). We placed this general schematic overview figure before the last paragraph of the introduction, where we describe the study because we think that it is helpful to state upfront who the subjects are, what timepoints are collected, how many samples there are, and what hypothesis are being addressed.

d) The intro is very long (the final paragraph is a full page).

Response: We shortened the introduction by removing parts that sounded redundant with the Results section.

5. The meta-analysis of RNA signatures and TB outcomes is not the best reference. Far richer and rigorous, and better validated and published signatures from MAs have been published, such as <https://www.ncbi.nlm.nih.gov/pmc/articles/PMC4838193/> and should rather be used. Furthermore, the Berry study is often incorrectly referred to as a meta-analysis in the manuscript text.

Response: Thank you for this comment, and we agree—in fact, this was a mistake on our part. We have updated the citation to reflect the meta-analysis that we were referring to from Anne O’Garra’s lab, where they re-analyzed the Berry cohort: <https://www.nature.com/articles/s41467-018-04579-w>. This paper is a meta-analysis where they derive a 373 gene signature. The paper referenced by the reviewer above is similar, however, that paper finds common genes across all datasets (3 gene TB signature), whereas the one we choose encompasses broader biology. We have updated the references to this throughout and updated the language to refer to this as the meta-analysis.

6. Results

a) Line 202: Why is Shannon diversity mentioned here?

Response: Thank you—this was a typo and is now updated to say “Inverse Simpson diversity”

b) Line 348-353: Please cite exact p-values for comparisons between time points in Figure 5A and B.

Response: We added the p values as requested.

c) Line 466: 18 FC and 36 CC do not add up to 52.

Response: We apologize and realize we actually have 55 people (19 and 36).

d) Rephrase using more formal language: “While these pathways ..., we do feel that ... disease” in lines 473-5 and “microbiome space” in line 478.

Response: We have updated the sentence to read more formally.

e) “Intestinal microbiome”, “gastrointestinal microbiome”, “gut microbiota” are all used

interchangeably. Authors should stick to one term throughout. More accurately, they measured the stool microbiome – not the (gastrointestinal) microbiome. The title should reflect this.

Response: We agree that we should be consistent and decided to go with gastrointestinal microbiota. We disagree with using stool microbiome, because there are plenty of studies showing that the stool microbiota resembles very closely the microbiota in the colon. Moreover, especially in clinical studies where only access to the microbiota is stool sampling it is commonly rereferred as GI/Gut/Intestinal microbiota.

7. Figures

Figure 1

a. Rather add an extra column for TTP in B and remove A and C as these sections do not contain information not already presented in B.

Response: We feel that the layout of Figure 1 (as it is) provides a nice overview of the study design, sample collection scheme, availability of samples, and questions that the paper attempt to answer by linking the different data modality. Hence, we decided to keep it (apart from specific edits addressed below).

b. In the legend for (A), indicate what HRZE and NTZ stand for. “TB-negative” should be changed to something more specific (i.e. IGRA-negative). In (C), the word “causal” is overly ambitious.

Response: We edited accordingly.

Figure 2

a. Fix spelling for “pretreatment” in (A).

Response: We fixed the typo.

b. Colours for HRZE and NTZ are difficult to distinguish.

Response: We changed the color throughout the paper to better distinguish.

c. In (B), quote exact p-values and state TTP is measured in hours on y-axis.

Response: We edited accordingly.

d. Annotate three symbols at bottom right of (C). and include PERMANOVA results.

Response: We edited accordingly.

e. X-axes in (B) and (D) should be only include days 0 and 14.

Response: We edited accordingly.

Figure 3

a. The colours in the volcano plot are difficult to distinguish.

Response: We agree that the colors may be difficult to distinguish. However, our rationale is that, throughout the paper, we are using unique and distinguishable colors for every Phylum and for each of these Phylum-specific colors we use different shades for every Order that belongs to that Phylum. Additionally, we have generated Supplementary Tables reporting the taxonomy, exact p-values and fold-change for all the ASVs displayed in Figure 3 (see Supplementary Data).

b. It would be beneficial to the reader to annotate the most differentially abundant taxa - at minimum those from the order Clostridiales in Figure 3A.

Response: We followed the reviewer suggestion and highlighted in Figure 3A those ASVs with highest Log₂ Fold Change values in each arm. For more information we redirect reviewers and readers to the Supplementary Tables reporting the taxonomy, exact p-values and fold-change for all the ASVs displayed in Figure 3.

c. Indicate clearly on the plot which timepoint is represented by a log-fold change <0 and which is >0. This applies to all volcano plots in the manuscript.

Response: We explicitly state in the caption that the Log₂ Fold Change is between day 14 (numerator) and day 0 (denominator) for the clinical trial cohort for both HRZE and NTZ arms of the trial.

Figure 4

a. Indicate what “NES” stands for.

Response: We now report in Figure 4 caption what NES stands for.

Figure 6

a. “Blue/orange entries indicate features found to significantly associate with changes in a specific inflammatory pathway,” however, the strength of this association (or statistical significance i.e. p-value or correlation coefficient with CI) is not described.

Response: We apologize if this was not clear in the text. All orange/blue cells in the heatmap are significant according to our permuted importance analysis calculations (FDR < 0.05). White cells indicate a lack of significance (e.g. empty). To address the reviewer’s concern, we now provide a dedicated table in the supplementary data that provide for every association in Figure 6 the variable importance value (i.e., the increase in mean square error of prediction accuracy if feature is removed), the slope and intercept from the linear approximation of the ALE curves and the related p values. We have added text to the Figure Caption redirecting readers to this Supplementary Data.

b. Similarly, can this be expressed statistically for “Black dots are used to identify the top important predictor”?

Response: Our goal with the black symbol is just to qualitatively show what is the top predictor found by the model for every immune pathway. That information can again be obtained by inspecting the provided Supplementary Data. We reworded the text in the caption of Figure 6 to explicitly say how the top predictor was determined.

Figure 7

a. Delete the colourful legend to the right in (A) - it is redundant with the top labels.

Response: We edited accordingly.

b. Groups should be arranged according to respective cohorts (i.e. EBA groups alongside each other)

Response: We arranged panels as suggested.

8. Methods

a) Were study participants on any antibiotics prior to study enrolment? A trial by antibiotics is common in many settings prior to patients commencing TB treatment.

Response: There was no formal antibiotic trial before HRZE treatment with active TB in this setting. We state this in the text under the “Donor recruitment and protection of human subjects” section of the Methods.

b) Were family contacts and community controls included in the study if both asymptomatic and IGRA-negative?

Response: Family contacts were enrolled if a TB case came to GHESKIO for care, at which point family members were asked. Family contacts and community controls could be both IGRA+ and IGRA-, however, these individuals did not have TB symptoms.

REVIEWERS' COMMENTS

Reviewer #3 (Remarks to the Author):

Dear Authors,

once again I apologize for the time it took me to review your work. I really try to do my reviewing work conscientiously and the time it requires needs to be stolen from my other duties.

After reading all our communications and the latest manuscript version, I am happy to say that I have no issues remaining, except to praise the authors for their persistence and patience with the somewhat intense reviewing process, as well as for the final result, which I believe is really good.

Reviewer #4 (Remarks to the Author):

In the revised version of the manuscript, the authors have addressed all of my remaining concerns. I accept the manuscript in its revised form.

Reviewer #5 (Remarks to the Author):

I thank the authors for their responses, which they have addressed well within the constraints of their data.

To respond further to one outstanding issue: my comment 1b) re the limitations of healthy controls

- Yes, being a completely independent cohort is good, no argument there
- Rather, one limitation is that this cohort, albeit independent, is in a different type of patient (healthy and not sick). Therefore, there is a risk that the signature they "validated" has overly optimistic specificity. This phenomenon is well documented in the diagnostic literature (e.g., <https://academic.oup.com/clinchem/article/51/8/1335/5629913>, "Individuals who do not have the target condition but who suffer from other diseases can be expected to produce false-positive results more often than otherwise healthy individuals."), perhaps not appreciated enough in the microbiome field, and requires clearer acknowledgement in the article text. An independent TB treatment cohort could be used, or even sick patients with other inflammation associated diseases (for which I am sure some datasets are public). The authors just need to highlight this for the reader and cool their language.

Response to comments for NCOMMS-20-29848A resubmitted as NCOMMS-20-29848B

Reviewer's Comments:

Reviewer #3 (Remarks to the Author):

After reading all our communications and the latest manuscript version, I am happy to say that I have no issues remaining, except to praise the authors for their persistence and patience with the somewhat intense reviewing process, as well as for the final result, which I believe is really good.

Response: We greatly appreciate the time and the constructive criticism from the reviewer. We also completely understand that with this time things are significantly slower and want to remark our appreciation for the time taken to review and comment.

Reviewer #4 (Remarks to the Author):

In the revised version of the manuscript, the authors have addressed all of my remaining concerns. I accept the manuscript in its revised form.

Response: We greatly appreciate the time and the constructive criticism from the reviewer.

Reviewer #5 (Remarks to the Author):

I thank the authors for their responses, which they have addressed well within the constraints of their data. To respond further to one outstanding issue: my comment 1b) re the limitations of healthy controls

- Yes, being a completely independent cohort is good, no argument there
- Rather, one limitation is that this cohort, albeit independent, is in a different type of patient (healthy and not sick). Therefore, there is a risk that the signature they "validated" has overly optimistic specificity. This phenomenon is well documented in the diagnostic literature (e.g., <https://academic.oup.com/clinchem/article/51/8/1335/5629913>, "Individuals who do not have the target condition but who suffer from other diseases can be expected to produce false-positive results more often than otherwise healthy individuals."), perhaps not appreciated enough in the microbiome field, and requires clearer acknowledgement in the article text. An independent TB treatment cohort could be used, or even sick patients with other inflammation associated diseases (for which I am sure some datasets are public). The authors just need to highlight this for the reader and cool their language.

Response: We greatly appreciate the time and the constructive criticism from the reviewer. We agree with the reviewer comment and edited the Discussion section to acknowledge this point, see page 20 line 657-662.